# Pathogenic *PDE12* variants impair mitochondrial RNA processing causing neonatal mitochondrial disease

Lindsey Van Haute [1], Petra Páleníková [1,18], Jia Xin Tang[2,19], Pavel A Nash[1], Mariella T Simon [3], Angela Pyle[4], Monika Oláhová[2,4], Christopher A Powell [1], Pedro Rebelo-Guiomar [1,20], Alexander Stover[3], Michael Champion[5], Charulata Deshpande[6,7], Emma L Baple[8,9,10], Karen L Stals[8], Sian Ellard[8,9], Olivia Anselem[11], Clémence Molac[11], Giulia Petrilli[12], Laurence Loeuillet[12], Sarah Grotto[13], Tania Attie-Bitach[12,14], Jose E Abdenur[3,15], Robert W Taylor [2,16✉] & Michal Minczuk [1,17✉]

## Abstract

Pathogenic variants in either the mitochondrial or nuclear genomes are associated with a diverse group of human disorders characterized by impaired mitochondrial function. Within this group, an increasing number of families have been identified, where Mendelian genetic disorders implicate defective mitochondrial RNA biology. The *PDE12* gene encodes the poly(A)-specific exoribonuclease, involved in the quality control of mitochondrial non-coding RNAs. Here, we report that disease-causing *PDE12* variants in three unrelated families are associated with mitochondrial respiratory chain deficiencies and wide-ranging clinical presentations *in utero* and within the neonatal period, with muscle and brain involvement leading to marked cytochrome *c* oxidase (COX) deficiency in muscle and severe lactic acidosis. Whole exome sequencing of affected probands revealed novel, segregating bi-allelic missense *PDE12* variants affecting conserved residues. Patient-derived primary fibroblasts demonstrate diminished steady-state levels of PDE12 protein, whilst mitochondrial poly(A)-tail RNA sequencing (MPAT-Seq) revealed an accumulation of spuriously polyadenylated mitochondrial RNA, consistent with perturbed function of PDE12 protein. Our data suggest that PDE12 regulates mitochondrial RNA processing and its loss results in neurological and muscular phenotypes.

**Keywords** Exome Sequencing; Lactic Acidosis; Mitochondrial Disease; RNA Processing; tRNA
**Subject Categories** Genetics, Gene Therapy & Genetic Disease; Organelles

See also: C Yu et al

## Introduction

Mitochondria are cellular organelles essential for a variety of metabolic processes, but their primary function is the production of ATP via the oxidative phosphorylation (OXPHOS) pathway (Nunnari and Suomalainen, 2012). The regulation of mitochondrial function is under dual genetic control and requires the concerted action of both nuclear and mitochondrial-encoded proteins (Smith and Robinson, 2019). It is estimated that around 1200 nuclear-encoded genes contribute to mitochondrial function, including 79 structural subunits of the OXPHOS machinery, various OXPHOS assembly factors, and proteins involved in the maintenance and expression of mtDNA (Rath et al, 2021). Consequently, mitochondrial diseases, which are often multi-system and early-onset fatal diseases, can be caused by genetic defects in both mitochondrial DNA (mtDNA) or nuclear-encoded genes. The majority of mitochondrial disorders involve genes that encode structural components or regulatory factors of the OXPHOS system or mtDNA maintenance and expression machinery (Peter and Falkenberg, 2020; Rahman and Copeland, 2019).

[1]MRC Mitochondrial Biology Unit, University of Cambridge, Cambridge, UK. [2]Mitochondrial Research Group, Translational and Clinical Research Institute, Faculty of Medical Sciences, Newcastle University, Newcastle upon Tyne, UK. [3]CHOC Children's Division of Metabolic Disorders, Orange, CA, USA. [4]Department of Applied Sciences, Faculty of Health & Life Sciences, Northumbria University, Newcastle upon Tyne, UK. [5]Department of Children's Inherited Metabolic Diseases, Evelina London Children's Hospital, Guy's & St Thomas' Hospital NHS Foundation Trust, London, UK. [6]Manchester Centre for Genomic Medicine, Central Manchester University Hospitals NHS Foundation Trust, Manchester, UK. [7]Department of Clinical Genetics, Guy's Hospital, Guy's & St Thomas' Hospital NHS Foundation Trust, London, UK. [8]Genomics Laboratory, Royal Devon University Healthcare NHS Foundation Trust, Exeter, UK. [9]Department of Clinical and Biomedical Sciences, University of Exeter Medical School, Exeter, UK. [10]Peninsula Clinical Genetics Service, Royal Devon University Healthcare NHS Foundation Trust, Exeter, UK. [11]Maternité Port-Royal, Département de Gynécologie-Obstétrique, Hôpital Cochin Broca Hôtel-Dieu, APHP, Paris, France. [12]Service de Médecine Génomique des Maladies Rares, Hôpital Necker-Enfants Malades, APHP, Paris, France. [13]UF de Génétique Clinique, Centre de Référence Anomalies du Développement et Syndromes Malformatifs, Hôpital Trousseau, APHP, Paris, France. [14]INSERM UMR 1163, Imagine Institute, Genetics and Development of the Cerebral Cortex, Université Paris Cité, Paris, France. [15]University of California, Irvine, Department of Pediatrics, Irvine, CA, USA. [16]NHS Highly Specialised Service for Rare Mitochondrial Disorders, Newcastle upon Tyne Hospitals NHS Foundation Trust, Newcastle upon Tyne, UK. [17]Department of Clinical Neurosciences, University of Cambridge, Cambridge, UK. [18]Present address: Broad Institute of MIT and Harvard, Cambridge, MA, USA. [19]Present address: Department of NanoBiophotonics, Max Planck Institute for Multidisciplinary Sciences, Göttingen, Germany. [20]Present address: Department of Biochemistry, University of Cambridge, Tennis Court Road, CB1 2GA Cambridge, UK. ✉E-mail: robert.taylor@ncl.ac.uk; michal.minczuk@mrc-mbu.cam.ac.uk

MtDNA is a highly compact, double-stranded, circular molecule that lacks introns and contains only 37 genes. Transcription of the L-strand promoter (LSP) results in a precursor transcript encoding eight mt-tRNAs and one mt-mRNA (ND6), while transcription from the H-strand results in a polycistronic transcript that encodes both rRNAs, 14 mt-tRNAs and 10 mt-mRNAs, containing 12 open reading frames (ORFs) (Anderson et al, 1981). These two long polycistronic precursor transcripts need further processing to release the individual mt-mRNA, mt-rRNA and mt-tRNA molecules. All proteins involved in mammalian mt-RNA transcription and maturation are encoded in the nuclear genome and imported into the mitochondrial matrix upon translation on cytoplasmic ribosomes.

The mt-tRNA genes mark most of the junctions between mitochondrial protein- and rRNA-coding genes, and according to the tRNA punctuation model (Montoya et al, 1983; Ojala et al, 1981), the secondary structures of the tRNA sequences provide the signals for excision of the tRNAs by the RNase P complex (5'-end endonucleolytic cleavage) and the RNase Z activity of ELAC2 (3'-end cleavage) (Brzezniak et al, 2011; Holzmann et al, 2008), yielding most of the mt-tRNAs, mt-mRNAs and mt-rRNAs. The asymmetrical processing mechanism of H- and L-strand polycistronic transcripts was observed in studies investigating the effect of a pathogenic mtDNA variant (Xiao et al, 2020; Zhao et al, 2019). All mt-RNAs then undergo further maturation, including the addition of a CCA trinucleotide to the 3'-end of all newly synthesised mt-tRNAs by tRNA nucleotidyltransferase (TRNT1) (Nagaike et al, 2001). With the exception of MT-ND6, all mt-mRNAs undergo polyadenylation by a mitochondrial poly(A) polymerase (mtPAP) (Mercer et al, 2011; Nagaike et al, 2005; Tomecki et al, 2004). This post-transcriptional addition of adenines to the 3' end of mt-mRNAs is involved in regulating their stability and is essential for several mt-mRNAs that lack the complete stop codon (MT-ND1, MT-ND2, MT-ND3, MT-ND4, MT-CYTB, MT-COIII and MT-ATP6) (Rorbach and Minczuk, 2012). Several pathogenic variants in nuclear genes have been detected that affect mt-RNA processing (Van Haute et al, 2015). Also, processing defects of mt-RNA precursors due to pathogenic mtDNA variants have been described (Guan et al, 1998; Ji et al, 2021; Wang et al, 2011; Xiao et al, 2020; Zhao et al, 2019).

Previously, we and others have identified phosphodiesterase 12 (PDE12) as a mitochondrial exonuclease/endonuclease/phosphate (EEP) protein (Poulsen et al, 2011; Rorbach et al, 2011), demonstrating that recombinant PDE12 exhibits a 3'–5', poly(A)-specific exoribonuclease activity. In previous work, we also showed that PDE12 acts as a major factor for quality control of mitochondrial non-coding RNAs (Pearce et al, 2017) and that the lack of PDE12 results in spurious polyadenylation of the 3'-ends of mt-tRNA and mt-rRNA species. This aberrant polyadenylation of selected mt-tRNAs led to reduced levels of mt-tRNAs available for aminoacylation, resulting in stalling of mitoribosomes at the corresponding codons (Desai et al, 2020; Pearce et al, 2017).

In this present work, we report the identification of rare, damaging missense variants in the PDE12 gene (NM_177966.7) in four individuals from three unrelated families, including two foetal siblings, who presented with severe, early-onset mitochondrial disease. The functional consequences of all three homozygous PDE12 variants on the maturation of non-coding mt-RNA species and their downstream impact on mitochondrial OXPHOS machinery were characterised in patient-derived fibroblasts, confirming that disruption of PDE12 function leads to disease pathology and marked mitochondrial biochemical deficiencies.

# Results

## Summary of clinical features of the investigated patients

Patient 1 is the first child, born to consanguineous, first-cousin parents. The patient died at 3 months of life. The couple's second child, Patient 2, required ventilation in the neonatal ICU and had persistent lactic acidosis. A developmental delay of 18 months was indicated at three and a half years old. Diagnostic muscle biopsy showed a mosaic pattern of cytochrome c oxidase (COX) histochemical activity (~60% COX-deficient fibres) and evidence of a combined CI + CIV defect following the direct assessment of respiratory chain enzyme activities. This child gradually improved and is currently 7 years old.

Patient 3 was born at 39 weeks via C-section to non-consanguineous parents. Prenatal ultrasound detected brain anomalies. At birth, the patient was limp, apnoeic and presented with respiratory failure and metabolic acidosis, requiring intubation. MRI of the brain was striking for lissencephaly, dysgenesis of the corpus callosum and extensive periventricular and subcortical cysts (Fig. 1). On day 2, life support was discontinued, and the patient died.

Foetus 4 was the second pregnancy of consanguineous parents who already had a healthy son. Prenatal ultrasounds first revealed increased nuchal translucency, later severe intra-uterine growth retardation, hydrops and cystic hygroma. This pregnancy ended spontaneously at 22 gestational weeks (Fig. EV1A–F). At 13 weeks of the third pregnancy of this couple (Foetus 5), a foetal ultrasound revealed nuchal translucency (14 mm) and the absence of foetal movements (Fig. EV1G–K). This pregnancy was terminated at 13 + 6 gestational weeks. Post-mortem examination was performed in both, and further clinical details for all patients can be found in the Appendix Supplementary Information. Table 1 provides a summary of the key clinical findings for all patients.

## Identification of PDE12 variants

Whole-exome sequencing was performed to identify the genetic cause of the observed clinical phenotypes (Fig. 2A), revealing different homozygous variants in the PDE12 gene (ENSG00000174840 / NM_177966.6, NP_808881.3) in all patients (Fig. 2B). Patient 1 and 2 have a homozygous missense variant in exon 1 (c.464 A > G, p.Tyr155Cys), WES in Patient 3 revealed a homozygous variant in exon 1 (c.1115 G > A, p.Gly372Glu), while foetal individuals 4 and 5 carried a homozygous c.122 G > C, p.Arg41Pro variant located in the mitochondrial targeting sequence (MTS) of PDE12 (Fig. 2B). All three missense variants alter evolutionarily conserved amino acids (Fig. 2C).

Based on the gnomAD database (https://gnomad.broadinstitute.org/), all three missense variants are rare within the general population with minor allele frequencies of less than 0.00001. Furthermore, three different in silico pathogenicity prediction tools pointed towards a high likelihood of pathogenicity for all three homozygous missense variants identified in the patient cohort. These were indicated as CADD scores of ≥20, 'probably damaging' on PolyPhen-2 and 'damaging' on SIFT.

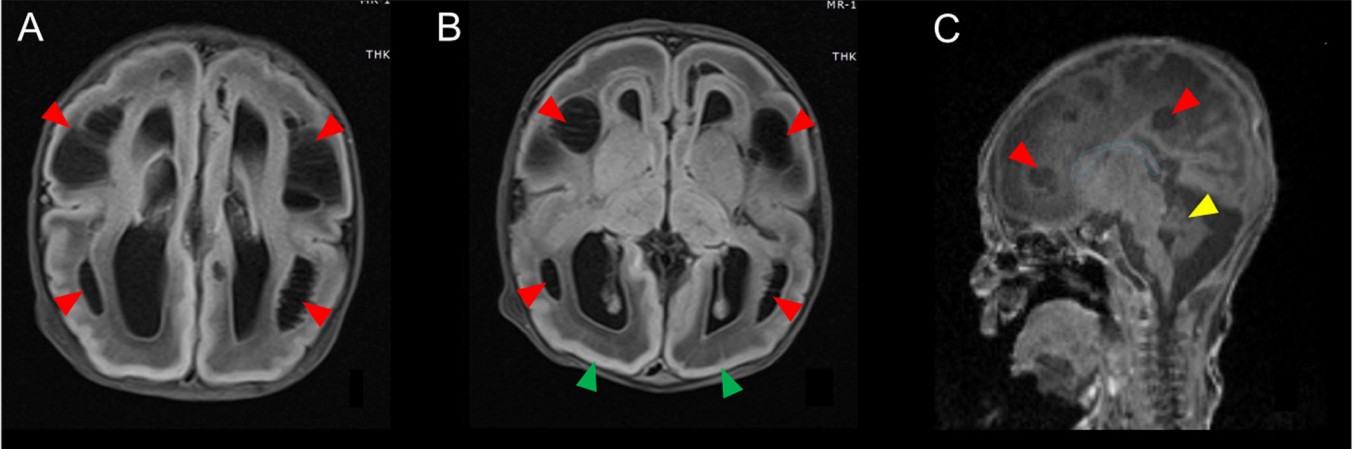

**Figure 1. Magnetic resonance imaging (MRI) of the brain of Patient 3.**

(A, B) Axial T2-flair. Extensive periventricular and subcortical cysts (red arrows) and lissencephaly (green arrows). (C) Sagittal T1. Severe dysgenesis of the corpus callosum (blue line) and small cerebellum (yellow arrow). Periventricular cysts are again demonstrated (red arrows). A brain MRI of Patient 3 was obtained on a 3-Tesla instrument.

**Table 1. PDE12 families in the study.**

|  | Patient 1 (Family 1) | Patient 2 (Family 1) | Patient 3 (Family 2) | Patient 4 (Family 3) | Patient 5 (Family 3) |
|---|---|---|---|---|---|
| Inheritance | Recessive | Recessive | Recessive | Recessive | Recessive |
| Gender | F | M | M | M | unknown |
| cDNA[a] | c.464 A > G | c.464 A > G | c.1115 G > A | c.122 G > C | c.122 G > C |
| Protein[a] | p.Tyr155Cys | p.Tyr155Cys | p.Gly372Glu | p.Arg41Pro | p.Arg41Pro |
| Allele Frequency in gnomAD[b] | 0.0000012 | 0.0000012 | 0.00000159 | 0.00000124 | 0.00000124 |
| Variant effect predictor | missense | missense | missense | missense | missense |
| Location | exon 1 | exon 1 | exon 1 | MTS | MTS |
| Variant type | homozygous | homozygous | homozygous | homozygous | homozygous |
| Disease evolution | Died at 3 months of life | Currently 7 years old | Died at 2 days of life | died in utero | termination of pregnancy |
| Consanguinity | yes | yes | no | yes | yes |

[a]Variants described according to GRCh38 and NM_177966.7.
[b]Heterozygous allele frequency taken from gnomAD v4.0.0; no homozygous occurrences of any variant.

## In silico characterisation of *PDE12* variants

Next, we aimed to predict the effect of the detected variants on PDE12 protein function. We used a published structural model of the protein to map the variants (Kim et al, 2015). The Tyr155 residue is located in one of the β-strands of the N-terminal domain of PDE12 (Fig. 2D). While Tyrosine has a bulky, aromatic side chain, Cysteine has a small, polar side chain, which may disrupt the hydrophobic environment of the core of this domain, contributed to by residues of neighbouring strands (e.g. F53, L148, I150) and amphipathic α-helix (V74, I78, A82) (Fig. 2E). The Tyr155Cys substitution may generate a reactive free thiol group; however, predicting the functional impact of this change is challenging, as Cys155 is only minimally exposed. Gly372 residue is located in a solvent-exposed region of one of the two beta-sheets of the catalytic domain, but it is not close to the catalytic site. Nearby Gly372, Phe357 and Phe385 interact via pi-stacking, potentially contributing to stabilising the domain (Fig. 2D,F). A change of Gly372 to

Glutamate would introduce a long side chain in the region, which could disrupt that hydrophobic interaction, perturbing the structure of the beta-sheet, ultimately disrupting the stability of the catalytic domain, and PDE12 itself. The predicted length of the mitochondrial targeting sequence (MTS) is 42 aa (Fig. 2B) (Kitada et al, 2003), consequently Arg41 could be involved in the mitochondrial processing peptidase (MPP) processing during protein import into the mitochondria. In summary, based on in silico analyses, the prediction of pathogenicity of the detected *PDE12* variants is challenging and, therefore, we set out to investigate their effects in the following in vitro and cellular studies.

## Patient *PDE12* variants affect protein stability, mitochondrial import and the 3′-end processing of mitochondrial RNAs

In order to investigate the functional consequences of the patient variants, we first determined steady-state PDE12 protein levels in

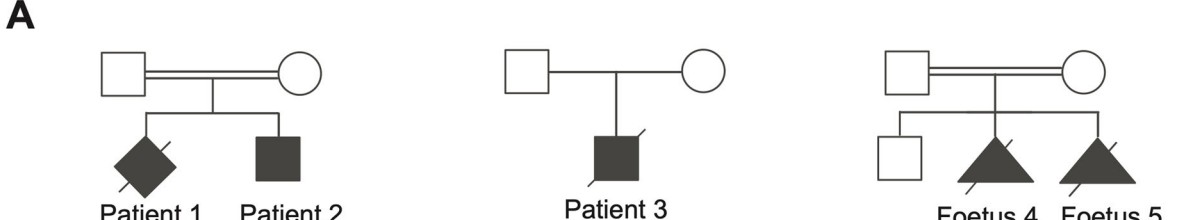

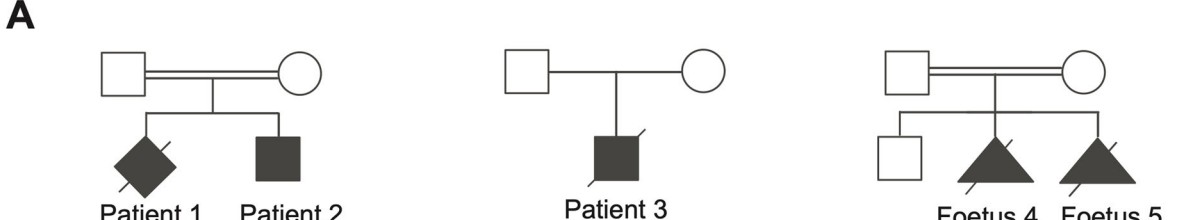

patient-derived fibroblasts. Cells from both Patient 2 and Patient 3 showed substantially diminished levels of PDE12 protein (Fig. 3A,B), consistent with the p.Tyr155Cys and p.Gly372Glu substitutions affecting protein stability. There was insufficient material from Foetus 4 and 5 to verify the PDE12 protein steady-state levels. To further investigate the potential impact of the p.Arg41Pro variant on MTS processing by MPP during mitochondrial import, we analysed the abundance of both pre- and processed protein using western blotting. For this purpose, we transiently overexpressed flag-tagged versions of either wild-type or p.Arg41Pro PDE12 in HeLa cells. Western blot analysis revealed that in cells overexpressing p.Arg41Pro PDE12 only unprocessed protein was detected, whereas

◀ **Figure 2. PDE12 gene structure and variants.**

(A) Pedigrees of the affected families. (B) Gene structure of *PDE12* and localisation of amino acid residues affected by mutations. The MTS (mitochondrial targeting sequence) and the Endonuclease/Exonuclease/phosphatase domain are indicated. (C) Conservation of human PDE12 amino acid residues affected by mutations across *Homo sapiens, Pan troglodytes, Rattus norvegicus, Mus musculus, Bos taurus, Canis familiaris, Xenopus tropicalis* and *Danio rerio*. (D) Overview of the structure of PDE12 (Q6L8Q7, AlphaFold prediction (Jumper et al, 2021), residues 154–609, with the substituted residues shown in red (Tyr155) and light red (Gly372). Magnesium ions in the catalytic site were obtained from PDB 4ZKF (Kim et al, 2015) and are presented as grey spheres. (E) Detailed view of the microenvironment of Tyr155, with a focus on neighbouring amino acids that contribute to the hydrophobic core of the N-terminal domain. (F) Detailed view of the microenvironment of Gly372, with a focus on amino acids with hydrophobic side chains.

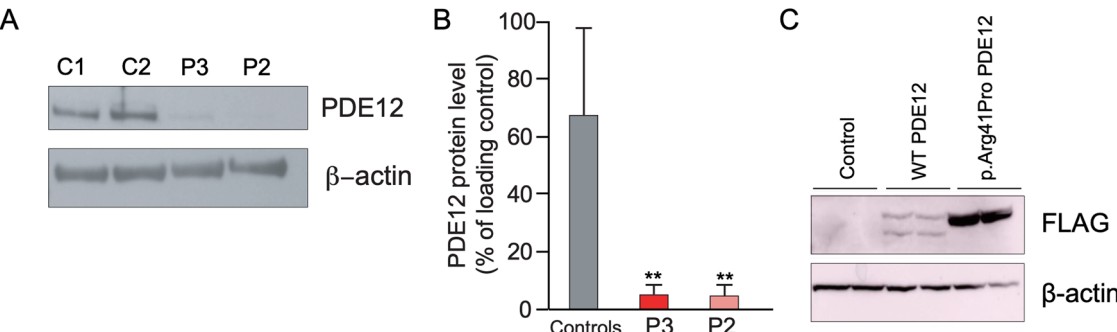

**Figure 3. Steady-state levels of PDE12 in patient cells.**

(A) Western blot detecting the PDE12 protein in human dermal fibroblasts from patients 2 and 3 and healthy controls. Beta-actin was used as a loading control ($n = 2$ technical replicates). (B) Quantification of PDE12 protein levels shown in (A) (**$p < 0.01$, control fibroblasts vs Patient 2: $p = 0.0012$, control fibroblasts vs Patient 3: $p = 0.0012$) ($n = 2$ technical replicates). (C) Western blot analysis of untransfected HeLa cells and cells transiently overexpressing wild-type or p.Arg41Pro PDE12-FLAG protein. Beta-actin was used as a loading control ($n = 2$ biological replicates). Source data are available online for this figure.

those overexpressing wild-type PDE12 showed both processed and unprocessed proteins (Fig. 3C). This finding strongly suggests that the p.Arg41Pro substitution, identified in foetal individuals 4 and 5, impairs processing by MPP and consequently protein import into the mitochondria.

We previously showed that PDE12 is necessary for the removal of spurious 3′ adenylation of mitochondrial RNA (Pearce et al, 2017). In these experiments, the lack of PDE12 in knockout (KO) HEK293T cells caused aberrant adenylation of a subset of mitochondrial tRNAs and rRNA. Therefore, we asked if a similar effect could be observed in patient-derived cells exhibiting lower levels of PDE12 protein. To this end, we analysed RNA isolated from Patient 2 and Patient 3 fibroblasts (Fig. 4) or foetal liver from Foetus 4 (Fig. 5) in the MPAT-Seq assay optimised to detect 3′ extensions in mitochondrial non-coding RNA transcripts (Fig. 4A) (Pearce et al, 2017). Compared to control fibroblasts, we detected a greater degree of polyadenylation of the 3′ end of the selected mt-tRNAs in Patient 2 and Patient 3 fibroblasts (Fig. 4B,C). We also compared the levels of spurious mt-tRNA polyadenylation in *PDE12* patient cells and HEK293T PDE12 KO cells. The poly(A) effect was stronger for mt-tRNA$^{Glu}$ (*MT-TE*) and mt-tRNA$^{Met}$ (*MT-TM*) compared with those in HEK293T PDE12 KO cells, but weaker for mt-tRNA$^{Lys}$ (*MT-TK*), which has the strongest aberrant adenylation in the HEK293T PDE12 KO. Apart from the effect on mt-tRNAs, we also observed aberrant adenylation of 16S rRNA in *PDE12* patient fibroblasts compared with controls, similar to the effect seen in HEK293T PDE12 KO cells (Fig. EV2A). Next, we performed MPAT-Seq using liver tissue from Foetus 4 and age-matched foetal control liver (Fig. 5). In this patient, the strongest effect on spurious mt-tRNA polyadenylation was seen for mt-

tRNA$^{His}$ (*MT-TH*) and mt-tRNA$^{Lys}$ (*MT-TK*) (Fig. 5A), which was about 10-fold greater than in controls (Fig. 5B). No substantial changes were detected for the 16S mt-rRNA 3′end extensions between Foetus 4 liver samples and controls (Fig. EV2B). Taken together, we conclude that the *PDE12* missense variants lead to decreased PDE12 protein levels and accumulation of dysfunctional, polyadenylated mt-tRNAs and, as a consequence, lower levels of mature mt-tRNAs available for aminoacylation.

## Analysis of OXPHOS protein levels and OXPHOS function in affected individuals

Our previous results showed that the absence of the PDE12 protein in HEK293T cells led to a marked loss in steady-state levels of OXPHOS proteins of complexes I, III and IV (Pearce et al, 2017). Accordingly, we asked if a similar effect could be detected in patient-derived cells harbouring *PDE12* variants (Figs. 2, 3). First, we used Blue-native polyacrylamide gel electrophoresis (BN-PAGE) in Patient 2 and Patient 3 fibroblasts, and PDE12 KO HEK293T cells as controls, to evaluate the integrity of all OXPHOS components. While HEK293T PDE12 KO cells clearly showed a marked defect in the integrity of OXPHOS complexes I, III and IV, compared to controls, this effect was not observed in fibroblasts derived from Patients 2 and 3 (Fig. 6A–C). Correspondingly, SDS–PAGE western blot analysis of individual subunits of OXPHOS complexes in Patient 2 and Patient 3 fibroblasts did not reveal any substantial changes compared to controls, apart from a mild MT-CO2 decrease in Patient 3 (Fig. 6D,E). Next, we measured basal and maximal oxygen consumption rate (OCR) (Fig. EV3A) and extracellular acidification rate (ECAR) (Fig. EV3E) in fibroblasts from *PDE12* patients 2 and 3 and controls.

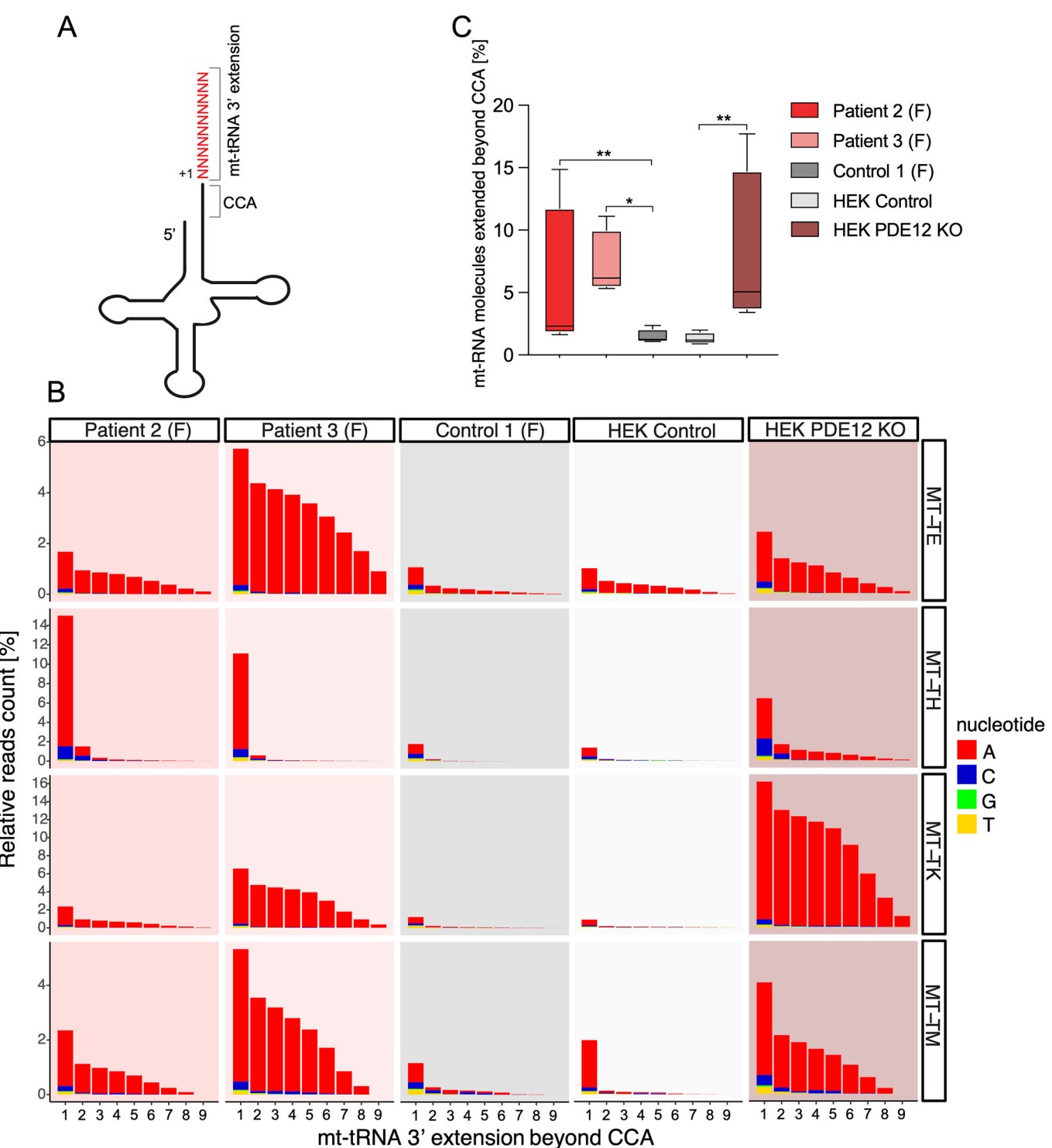

**Figure 4.  Spurious 3′ adenylation in the absence of PDE12 in patient fibroblasts.**

(**A**) Schematic of mt-tRNA structure depicting 3′ extension determined via the MPAT-Seq assay. (**B**) Representation of 3′ ends of a subset of mt-tRNAs from patient 2 and patient 3 fibroblasts "(F)", age-match fibroblast controls, HEK293T PDE12 KO cells and HEK293T control cells, ascertained by MPAT-Seq. The read count shown for each position is relative to the read count for the last A in CCA for each mt-tRNA (MT-TE (mt-tRNA$^{Glu}$), MT-TH (mt-tRNA$^{His}$), MT-TK (mt-tRNA$^{Lys}$), MT-TM (mt-tRNA$^{Met}$)). (**C**) Box and whisker plot representation of the percentage of mt-tRNAs molecules extended beyond the 3′ CCA addition for the analyzed mt-tRNAs shown in (**B**) ($n = 1$ biological replicate). The two-tailored $t$-test was used to calculate the $p$ values (control fibroblasts vs Patient 2: $p = 0.0015$, control fibroblasts vs Patient 3: $p = 0.0202$, HEK293T PDE12 KO vs HEK293T control cells: $p = 0.0025$), 4 mt-tRNAs were analysed with $n = 1$. Whiskers show min and max values, and the line in the centre represents the median. Source data are available online for this figure.

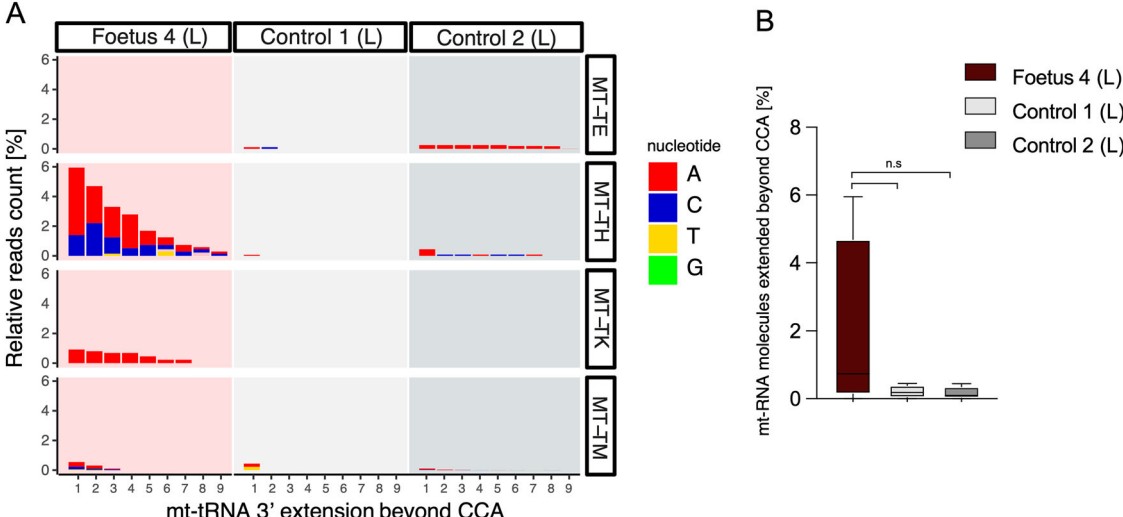

**Figure 5. Spurious 3′ adenylation in the absence of PDE12 in patient liver.**

(A) Representation of 3′ ends of a subset of mt-tRNAs extracted from Foetus 4 liver "(L)" tissue and two age-matched control livers, analysed by MPAT-Seq. The read count shown for each position is relative to the read count for the last encoded nucleotide for each mt-tRNA. (B) Box and whisker plot representation of the percentage of mt-tRNAs molecules extended beyond the 3′ CCA addition for the analyzed mt-tRNAs shown in (A). A two-tailored $t$-test was used to calculate the $p$ values ($p = 0.28$), and 4 mt-tRNAs were analysed with $n = 1$. Whiskers show min and max values, and the line in the centre represents the median. Source data are available online for this figure.

While no changes in OCR were observed in the patient cells compared with controls (Fig. EVB–D), ECAR measurements revealed a marginally increased basal glycolysis level in patient fibroblasts (Fig. EV3F). Given no detectable effect on OXPHOS complex steady-state levels present in Patient 2 and Patient 3 fibroblasts, we next analysed available skeletal muscle samples from Patient 2. BN-PAGE analysis showed a marked decrease in complex I (NDUFB8) and complex IV (MT-CO1) integrity in skeletal muscle, while the assembly of complex III (UQCRC2) was only mildly decreased compared to controls (Fig. 6F,G). The steady-state level of Complex II, the only OXPHOS complex exclusively encoded by the nuclear genome, was slightly increased in Patient 2 skeletal muscle (Fig. 6F,G). While a muscle biopsy from Patient 3 was unavailable, we analysed foetal skeletal muscle from Foetus 4 (Fig. 6H, I) by western blotting. We could not detect ATP5A (Complex V), MT-CO2 (Complex IV) and NDUFB8 (Complex I), the steady-state level of UQCRC2, a protein from complex III, was strongly decreased, while only a slight reduction of SDHB (Complex II) was observed (Fig. 6H,I), suggesting dramatic perturbance on the stability of OXPHOS subunits in the PDE12 Foetus 4 skeletal muscle. Taken together, these results indicate that p.Tyr155Cys (Patient 2) and p.Arg41Pro (Foetus 4) substantially compromise OXPHOS biogenesis and function in patient skeletal muscle, while the remaining levels of p.Tyr155Cys (Patient 2) or p.Gly372Glu (Patient 3) PDE12 can sustain OXPHOS in patient fibroblasts.

### Pathway analysis in affected individuals

Finally, in order to analyse perturbations in specific cellular pathways contributing to pathogenesis in our patient cohort, we used Affymetrix Clariom D transcriptome arrays to study the RNA levels in fibroblasts derived from both Patient 2 and Patient 3,

comparing these to age-matched controls. We used Metascape (Zhou et al, 2019) to perform system-level analysis (Fig. EV4A,B). Strikingly, most of the differentially expressed pathways affect growth and tissue development, such as tube morphogenesis and heart development (Fig. EV4A). This could potentially explain the very early phenotype, as observed by prenatal ultrasounds of Foetuses 4 and 5.

## Discussion

In the present study, we describe five patients, from three unrelated families, with homozygous *PDE12* missense variants. Previously, we showed that PDE12 is a key factor involved in the maturation and quality control of mitochondrial non-coding RNAs. The lack of PDE12 in human cultured cells results in a spurious polyadenylation of the 3′ ends of mt-rRNA and mt-tRNA, causing reduced levels of mt-tRNAs available for aminoacylation and stalling of mitoribosomes (Desai et al, 2020; Pearce et al, 2017).

In this study, we show that rare *PDE12* variants may cause an early-onset mitochondrial disease presentation, with the first symptoms observed during development in utero. We show that diminished levels of the PDE12 protein in patient skeletal muscle result in decreased steady-state levels of OXPHOS proteins. In the foetal skeletal muscle tissue of Foetus 4, the impact on OXPHOS protein steady-state levels were particularly profound with NDUFB8, MT-CO1 and ATP5A which constitutes complex I, IV and V of the OXPHOS system, respectively, being undetectable in the applied conditions (Fig. 6). This suggests that the c.122 G > C, p.Arg41Pro variant has a more severe effect than the other variants, namely c.464 A > G p.Tyr155Cys and c.1115 G > A p.Gly372Glu. Given that the predicted length of the MTS in PDE12 is 42 aa, we investigated whether Arg41 plays a role in

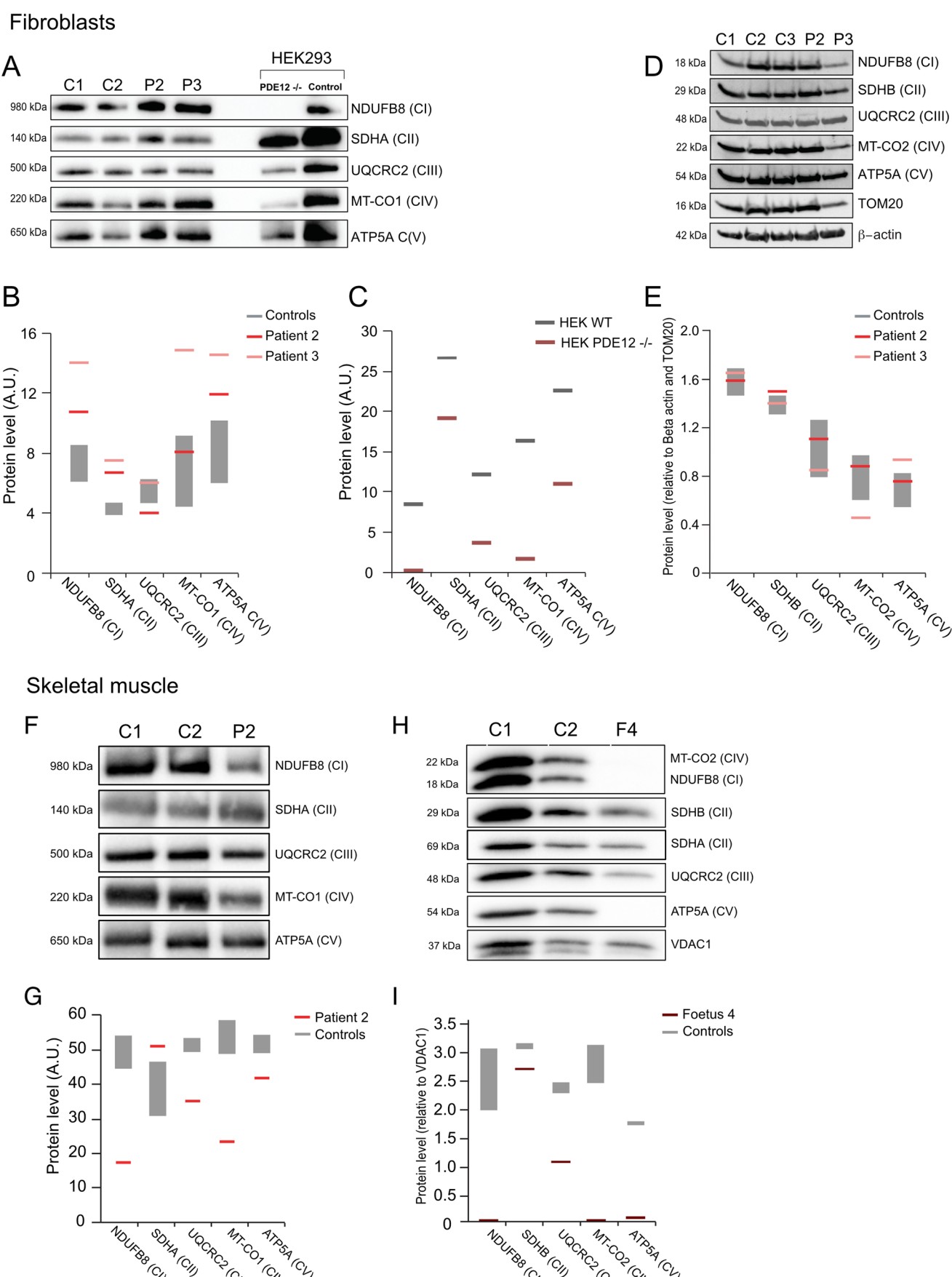

**Figure 6. The effect of *PDE12* variants on OXPHOS components in patient fibroblasts and skeletal muscle.**

(A) BN-PAGE analysis for NDUFB8 (complex I), SDHA (complex II), UQCRC2 (complex III), MT-CO1 (complex IV) and ATP5A (complex V) from *PDE12* patients (Patient 2, Patient 3) fibroblasts, control fibroblasts (C1, C2) and HEK293T cells lacking a functional PDE12 and HEK293T control cells ($n = 1$). (B) Quantification of (E) for protein extracts from fibroblasts. Grey boxes represent the spread of the control fibroblast analysis. (C) Quantification of (E) protein extracts from HEK293T cell lines. (D) Western blot analysis of NDUFB8 (complex I), SDHB (complex II), UQCRC2 (complex III), MT-CO2 (complex IV) and ATP5A (complex V) for *PDE12* patients and control fibroblasts. TOM20 and Beta-actin were used as loading controls. ($n = 1$). (E) Quantification of (D). Grey boxes represent the spread of the control fibroblasts. (F) BN-PAGE analysis for NDUFB8 (complex I), SDHA (complex II), UQCRC2 (complex III), MT-CO1 (complex IV) and ATP5A (complex V) for skeletal muscle of *PDE12* patient 2 and wild-type controls ($n = 1$). (G) Quantification of (A). The grey bars show the spread of the two controls. (H) Western blot analysis of Foetus 4 (F4) skeletal muscle and wild-type controls for ATP5A (complex V), UQCRC2 (complex III), SDHB (complex II), MT-CO2 (complex IV) and NDUFB8 (complex I). VDAC1 and SDHA (complex II) were used as loading controls ($n = 1$). (I) Quantification of (C). The grey bars show the value range of both controls. Source data are available online for this figure.

maturation by mitochondrial processing peptidase. Our results demonstrated that the p.Arg41Pro variant impairs the mitochondrial import of PDE12, as evidenced by the absence of detectable protein processing when the protein with this alteration is overexpressed (Fig. 3C). It is plausible that Arg41Pro disrupts the recognition site for MPP, which is predicted to cleave between Cys42 and Val43 (Kitada et al, 2003). If the processing of the PDE12 MTS is similarly compromised in foetal individuals 4 and 5, this impairment could account for the severe phenotypes observed in these subjects. The other two variants have very low, yet detectable levels of PDE12, presumably resulting in a low, but existing efficiency of the remaining PDE12 protein.

Our previous study showed that knockout of PDE12 leads to increased mitochondrial ribosome stalling events, mainly at lysine codons (Desai et al, 2020; Pearce et al, 2017). The mitochondrial polyadenylation test (MPAT)-Seq revealed that some other mt-tRNAs were also affected (mt-tRNA^His (MT-TH), mt-tRNA^Ser(AGY) (MT-TS2)), while others remained largely unchanged (Pearce et al, 2017). The same study demonstrated that mitochondrial ribosomal 16S rRNA also harbours 3′ end oligoadenylation, without affecting mitochondrial ribosome assembly. In this study, we provide evidence that fibroblasts derived from Patient 2, who is currently 7 years old, only exhibit limited polyadenylation at the 3′ ends of selected mt-tRNAs and 16S rRNA (Figs. 4 and EV2A). Patient 3, who died following the discontinuation of life support two days after birth, shows much higher polyadenylation levels than Patient 2. Notably, certain mitochondrial tRNAs (mt-tRNA^Glu (MT-TE) and mt-tRNA^Met (MT-TM)) display even higher polyadenylation levels in Patient 3 than observed in the PDE12 knockout cells (Figs. 4 and EV2A). While fibroblasts from Foetus 4 were not available, we performed MPAT-Seq on liver tissue and compared this with foetal control tissue. In this tissue, mt-tRNA^His (MT-TH) was the most affected mt-tRNA (Figs. 5 and Fig EV2B). It is still unclear why certain mt-tRNAs seem to be more affected than others. In our previous study, where we used HEK293T knockout PDE12 cells, the two of the most affected tRNAs were (mt-tRNA^His (MT-TH) and mt-tRNA^Lys (MT-TK)) (Pearce et al, 2017). The same two mt-tRNAs are most affected in foetal individual 4, the only variant that severely disrupts mitochondrial import. This suggests that there might be a difference between the absence of functional PDE12 and low-level PDE12 function, such as in Patients 1–3.

Taken together, the biochemical findings in patient-derived biological samples aligned with the age-of-onset and prognosis of the patients with Foetus 4 harbouring the c.122 G > C p.(Arg41Pro) variant being most severely affected, followed by Patient 3 variant c.1115 G > A p.Gly372Glu, and the mildest of the three patients being present in Patient 2 (c.464 A > G p.Tyr155Cys). The results of

transcriptomic analyses also suggest the importance of wild-type PDE12 expression in embryonic development, corroborating the prenatal developmental defects observed for foetal individuals 4 and 5.

Certain mitochondrial diseases have been identified as reversible infantile conditions. Through extensive life-sustaining interventions, patients who survive the critical first year of life can undergo recovery and progress towards relatively normal development, possibly facilitated by mTOR activation and enhanced mitochondrial biogenesis (Hathazi et al, 2020). Examples are individuals with a pathogenic, homoplasmic *MT-TE* tRNA variant or with a defect in proteins involved in *MT-TE* biology, including mitochondrial Glutaminyl-tRNA Synthetase, EARS2, or mitochondrial tRNA^Glu-modifying enzyme TRMU. In this same study, the authors investigated potential genetic modifiers, identifying a recurrent missense variant (c.67 C > T, p.Arg23Trp) in the *PDE12* gene, which was present in the homozygous state in 8 out of 19 affected families that were studied, but only in one of the control families. While Patient 1 died at three months of age and Patient 2 needed intensive care soon after birth, Patient 2 is currently 6/7 years old and has improved slowly over time, indicating that a similar reversible effect might be responsible for this improvement. Further research into the role of PDE12 as a potential genetic modifier is therefore needed.

In summary, we report the identification of pathogenic variants in the nuclear-encoded phosphodiesterase 12 (*PDE12*) gene that underlies the phenotypes present in the three patients we have studied. Our results demonstrated novel homozygous missense variants in the *PDE12* gene that lead to spurious polyadenylation on the 3′ end of several non-coding mitochondrial RNAs resulting in combined OXPHOS deficiency.

## Methods

**Reagents and tools table**

| Reagent/resource | Reference or source | Identifier or catalogue number |
|---|---|---|
| **Experimental models** | | |
| PDE12 KO HEK293T | Pearce et al, 2017 | N/A |
| HeLa | ATCC | CCL-2 |
| **Recombinant DNA** | | |
| WT PDE12.Strep2.Flag plasmid | Rorbach et al, 2011 | N/A |
| **Antibodies** | | |
| anti-TOMM20 | Abcam | ab186735 |

| Reagent/resource | Reference or source | Identifier or catalogue number |
|---|---|---|
| Complex I subunit NDUFB8 | Abcam | ab110242 |
| Complex II subunit SDHB | Abcam | ab14714 |
| SDHA | Abcam | ab14715 |
| Complex III subunit Core 2 | Abcam | ab14745 |
| Complex IV subunit I | Abcam | ab91317 |
| ATP synthase subunit alpha | Abcam | ab14748 |
| PDE12 | Abcam | ab87738 |
| VDAC1 | Abcam | ab14734 |
| Flag M2 | Sigma-Aldrich | F1804 |
| Total OXPHOS Human WB Antibody Cocktail | Abcam | ab110411 |
| beta-actin | Sigma-Aldrich | A5441 |
| **Oligonucleotides and other sequence-based reagents** | | |
| Oligonucleotides | This study | Table 2 |
| **Chemicals, enzymes and other reagents** | | |
| T4 RNA ligase | NEB | M0437M |
| Turbo DNase | Thermo Fisher Scientific | AM2238 |
| RNase inhibitor | Thermo Fisher Scientific | N8080119 |
| Superscript II Reverse Transcriptase | Thermo Fisher Scientific | 18064014 |
| Phusion polymerase | NEB | M0530S |
| NexteraXT DNA library preparation kit | Illumina | FC-131-1024 |
| NheI | NEB | R3131S |
| AflII | NEB | R0520S |
| Gibson assembly kit | NEB | E5510S |
| FugeneHD | Promega | E2311 |
| Twist Human Core Exome Kit | Twist Bioscience | |
| High Glucose Dulbecco's Modified Eagle Media (DMEM) | Thermo Fisher Scientific | 11965092 |
| TRIzol™ reagent | Thermo Fisher Scientific | 15596026 |
| **Software** | | |
| R (4.1.1) | https://www.r-project.org/ | N/A |
| Rstudio | https://posit.co/download/rstudio-desktop/ | N/A |
| Transcriptome Analysis Console (TCA) software | Thermo Fisher Scientific | N/A |
| Illumina Local Run Manager | Illumina | N/A |
| Metascape | https://metascape.org | N/A |
| Combined Annotation Dependent Depletion (CADD) | https://cadd.gs.washington.edu/ | N/A |
| Polymorphism Phenotyping v2 (PolyPhen-2) | http://genetics.bwh.harvard.edu/pph2 | N/A |
| Sorting Intolerant From Tolerant (SIFT) | https://sift.bii.a-star.edu.sg | N/A |

| Reagent/resource | Reference or source | Identifier or catalogue number |
|---|---|---|
| GraphPad Prism 10 | GraphPad Software LLC | N/A |
| **Other** | | |
| SDS-PAGE 4–12% bis-tris gels | Life Technologies | NP0321BOX |
| iBlot 2 Dry Blotting System | Thermo Fisher Scientific | IB21001 |
| MiSeq / HiSeq / NovaSeq | Illumina | N/A |
| DH5a sub-cloning efficiency cells | Invitrogen | 18265017 |
| Seahorse XFe24 Analyzer | Agilent | N/A |
| PierceTM BCA Protein Assay Kit | Thermo Fisher Scientific | 23225 |
| Native PAGE™ 4–16% Bis-Tris 1.0 mm Mini Protein Gels | Life Technologies | BN2111BX10 |
| Agilent Clinical Research Exome kit | Agilent | N/A |
| Covaris S2 Ultrasonicator | Covaris | N/A |

## Ethics statement

Informed consent for diagnostic and research-based studies was obtained for all subjects in accordance with the Declaration of Helsinki protocols and approved by local institutional review boards. All clinical investigations were evaluated according to the Declaration of Helsinki, and the experiments conformed to the principles set out in the Department of Health and Human Services Belmont Report. Parental consent was obtained to publish the clinical photography without any restriction. Ethical approval was granted by the Newcastle and North Tyneside Research Ethics Committee (REC reference: 16/NE/0267). The genetic analysis and autopsy were performed as part of routine clinical diagnosis, with signed parental consent for both, in accordance with French law. The use of DNA samples for research has been approved by the Ile de France II Ethics Committee (N° 2009-164 of 25/01/2010) and is subject to a declaration of collection (DC-2011-1449).

## Diagnostic genomic testing

Trio exome (Agilent/Illumina) sequence analysis using DNA from Patient 1 and parents was performed as described previously (Chen et al, 2023). Variants were filtered to select rare coding and intronic (−50 to +10) non-synonymous variants by inheritance (de novo, autosomal recessive, compound heterozygous).

Whole-exome sequencing of Patient 2 was undertaken by enriching coding regions using Twist Human Core Exome Kit, followed by sequencing in 100 base pair paired-end reads on the NovaSeq™ 6000 Sequencing System. Bioinformatics filtering and prioritisation of candidate variants was conducted using an in-house bioinformatics pipeline as described in (Deen et al, 2023). In silico pathogenicity prediction tools were used to assess the likelihood of pathogenicity for the single nucleotide variants, namely combined annotation-dependent depletion (CADD; https://cadd.gs.washington.edu/), Polymorphism Phenotyping v2 (PolyPhen-2; http://genetics.bwh.harvard.edu/pph2/) and Sorting

**Table 2. Primers used to generate the plasmid expressing PDE12-Flag p.Arg41Pro.**

| Primer name | Sequence |
| --- | --- |
| Gibson Fwd | CTGAACAGGCCGCACCGCCGCTAGCATTCGGCCGGCT |
| Mut Fwd | TGGGTTCCGAAGGTACGCAGGGCACTACAGCGCGCTCCATC |
| Mut Rev | GATGGAGCGCGCTGTAGTGCCCTGCGTACCTTCGGAACCCA |
| Gibson Rev | CCGGACTCTAGCGTTTAAACTTAAGATGTGGAGGCTCCCAG |

Intolerant From Tolerant (SIFT; https://sift.bii.a-star.edu.sg/). The presence of the genetic variant was further confirmed using Sanger sequencing.

The whole exome sequence of Patient 3 was performed by Gene Dx (Gaithersburg, MD, USA) as a family trio (patient and both parents). Per the laboratory report, the Agilent Clinical Research Exome kit was used to target the exonic regions and flanking splice junctions of the genome. These targeted regions were sequenced simultaneously by massively parallel sequencing (Illumina HiSeq) with 100 bp paired-end reads. The bi-directional sequence was assembled, aligned to reference gene sequences based on human genome build GRCh37/UCSC hg19, and analysed for sequence variants using a custom-developed analysis tool (Xome Analyzer). Sequence alterations were reported according to the Human Genome Variation Society (HGVS) nomenclature guidelines. According to the phenotype information provided, the analysis specifically included a review of variants in genes associated with lactic acidosis, metabolic and mitochondrial disease/dysfunction, abnormalities of the corpus callosum, periventricular cysts, infantile encephalopathy, neonatal respiratory failure or hypotonia. Additionally, the mitochondrial DNA sequence in the proband did not identify any pathogenic variant.

For Patients 4 and 5, microarray chromosomal analysis found no pathogenic copy number variant. The genomic DNA was extracted from the foetal tissue of the probands and the peripheral blood of the parents. With signed parental consent for the genetic analysis, we performed whole-exome sequencing (WES) on the two affected siblings and their healthy parents. Briefly, Twist libraries were prepared from 3 μg of genomic DNA sheared with a Covaris S2 Ultrasonicator. Exome capture was performed with the Human Core Exome Kit (Twist Bioscience). Sequencing was carried out using a NovaSeq (Illumina). After demultiplexing, paired-end sequences were aligned to the reference human genome (NCBI build37/hg19 version) using Burrows-Wheeler Aligner for Illumina data. The mean depth of coverage obtained for each sample was greater than 140X, with more than 95% of the exome covered at least 30X. Downstream processing was carried out with the Genome Analysis Toolkit (GATK), SAMtools, and Picard (http://www.broadinstitute.org/gatk/guide/topic?name=best-practices). Variant calls were made with the GATK Unified Genotyper. In each case, an in-house software tool (PolyWeb) was used to annotate (based on Ensembl release 71) and filter variants according to relevant genetic models. We excluded known variants listed in the public databases and variants previously identified in "in-house" exomes variants. Then, we selected variants affecting splice sites or coding regions (non-synonymous substitutions, insertions or deletions). Sanger sequencing was performed to confirm variants and to establish phase.

## Cell lines and culture conditions

Primary fibroblasts derived from patients, age-matched controls, previously generated PDE12 KO HEK293T (Pearce et al, 2017) and HeLa (ATCC, CCL-2) cells were cultured in High Glucose Dulbecco's Modified Eagle Media (DMEM) supplemented with 10% (v/v) foetal bovine serum (FBS), 1X non-essential amino acids, 50 μg/ml penicillin, 50 μg/ml streptomycin and 50 μg/ml uridine. Cell cultures were maintained at 37 °C under a 5% $CO_2$ atmosphere. Cell cultures were regularly tested for mycoplasma contamination.

## RNA extraction HEK293T cells, liver and fibroblasts

RNA was extracted from fibroblasts and HEK293T cells with TRIzol™ reagent according to the manufacturer's instructions. For foetal liver biopsies, at least 1 mg of each sample was powdered in liquid nitrogen using a pestle and mortar prior to RNA extraction using a TRIzol™ reagent.

## Mitochondrial poly(A) tail sequencing (MPAT-Seq)

This method was performed as described previously by Pearce and colleagues (Pearce et al, 2017) with the following modifications. In brief, 2.5 μg of total RNA (200 ng for foetal liver) was incubated for 15 min at 4 °C with 40 pmol of oligonucleotide, which carries a 3′ Spacer C3 modification to prevent concatemerisation. Next, ligation was performed for 2 h at 25 °C using 10 U T4 RNA ligase (NEB), 1 mM adenosine triphosphate, 2 U Turbo DNase and 40 U RNase inhibitor) in 20 μl reactions. After phenol:chloroform extraction, ligated RNA was subjected to reverse-transcription for 60 min at 42 °C using Superscript II primed with 10 pmol of ANTI-LIGN oligonucleotide. 12.5% of this reaction was used as a template for PCR using 10 pmol of a gene-specific Fw1 primer and a reverse ANTI-LIGN primer with an overhang sequence compatible with the NexteraXT using Phusion polymerase. PCR products were pooled and used as templates to produce Illumina sequencing libraries with NexteraXT, which were subjected to high-throughput sequencing using the Illumina MiSeq platform (75PE).

## Computational analysis of MPAT-Seq data

Cutadapt (Martin, 2011) was used to remove the MPAT primer sequences from the reads and reads longer than 27 nt were mapped using Bowtie2 (Langmead and Salzberg, 2012). Reads were aligned to a reference genome based on GRCh38, but with 30 additional uncalled positions (N) on the 3′ end of all mitochondrial rRNAs and tRNAs, to ensure that all 3′ ends were mapped. To further improve the alignment of the 3′ ends, mapping parameters -rdg 7,5 and -rfg 7,5 np 0 were used.

## Immunodetection of proteins by western blotting

For immunoblotting analysis, 20–30 μg of extracted proteins were loaded on SDS-PAGE 4–12% bis-tris gels (Life Technologies) and transferred onto a membrane using an iBlot 2 Dry Blotting System (Thermo Fisher Scientific) or separated by SDS–PAGE (12% gel), followed by a wet transfer to polyvinyl difluoride (PVDF)

membrane or nitrocellulose membrane. The following antibodies were used: anti-TOMM20 (Abcam, ab186735), Complex I subunit NDUFB8 (Abcam, ab110242), Complex II subunit SDHB (Abcam, ab14714) or SDHA (Abcam, ab14715), Complex III subunit Core 2 (Abcam, ab14745), Complex IV subunit I (Abcam, ab91317) and ATP synthase subunit alpha (Abcam, ab14748), PDE12 (Abcam, ab87738), VDAC1 (Abcam, ab14734), Flag M2 (Sigma-Aldrich, F1804) Total OXPHOS Human WB Antibody Cocktail (ab110411) and beta-actin (Sigma-Aldrich, A5441). All antibodies were validated by the manufacturer.

### Generation and transient expression of PDE12 plasmids

The generation of the WT PDE12.Strep2.Flag plasmid was described in our previous work (Rorbach et al, 2011). To generate the plasmid encoding PDE12-FLAG p.Arg41Pro the WT PDE12.Strep2.Flag plasmid was first digested with NheI and AflII. The p.Arg41Pro change was then installed by site-directed mutagenesis and Gibson assembly using the NEB Gibson assembly master mix. The primer sequences can be found in Table 2. Phusion 2X polymerase was used for amplification. Clones were rescued with DH5α sub-cloning efficiency cells (Invitrogen). HeLa cells were transfected with 2 µg plasmids with FugeneHD (Promega) reagent using a 1:2.5 DNA: Fugene ratio in biological duplicate. Cells were left for 48 h prior to protein solubilisation with RIPA buffer.

### Assessment of mitochondrial respiration

Primary fibroblasts were seeded at 30,000 cells per well in Seahorse XFe24 microplates and incubated at 37 °C, 5% $CO_2$. After 18 h, the cell culture medium was replaced with Seahorse assay medium consisting of Seahorse XF DMEM, 15 mM L-glutamine, 0.5 mM sodium pyruvate and 5 mM glucose to equilibrate at 37 °C at atmospheric $CO_2$ for 1 h. Oxygen consumption rate (OCR) and extracellular acidification rate (ECAR) were measured using the Seahorse XFe24 Extracellular Flux Analyzer. Measurements were performed in the following order: (i) basal respiration, (ii) uncoupled mitochondrial respiration through the injection of 1 µM oligomycin, (iii) maximal respiration by injecting 1 µM FCCP and (iv) non-mitochondrial respiration using 0.5 µM rotenone and antimycin. The assay was conducted with at least three replicates.

### Blue-native polyacrylamide gel electrophoresis (BN-PAGE)

Frozen skeletal muscle samples were processed and resuspended in a homogenisation buffer containing 250 mM sucrose, 20 mM imidazole hydrochloride and 100 mM phenylmethanesulphonyl fluoride (PMSF). Subsequently, samples were subjected to 15–20 rounds of homogenisation using a Teflon glass Dounce homogeniser at 4 °C. Mitochondria were pelleted at 20,000 × g for 10 min at 4 °C and washed with the homogenisation buffer before the final centrifugation step at 20,000 × g for 5 min at 4 °C to remove excess supernatant. The mitochondrial protein-enriched pellets were then solubilised in 2% n-dodecyl β-D-maltoside (DDM) by incubating on ice for 20 min with intermittent dispersion using glass spatulas. Following ultracentrifugation at

---

**The paper explained**

**Problem**

The human body relies on ~1200 nuclear genes to produce proteins essential for the functioning of mitochondria. Mutations in any of these genes can lead to a wide variety of symptoms, which makes diagnosing mitochondrial disorders quite difficult at the molecular level.

**Results**

In our study, we discovered five patients from three unrelated families who have rare identical genetic changes in the *PDE12* gene. Our research showed that when PDE12 protein levels are severely diminished, this disrupts the normal processing of mitochondrial RNA. This disruption causes errors in the RNA, particularly affecting its polyadenylation in mitochondria, and leads to instability in crucial components of the OXPHOS system. These issues primarily result in severe lactic acidosis, which particularly impacts muscle and brain function.

**Impact**

We have shown that normal levels of PDE12 are vital for proper embryonic development. Variants in the *PDE12* gene can cause a mitochondrial disease that begins in infancy and results in combined deficiencies in the OXPHOS system.

---

100,000 × g for 15 min at 4 °C, the supernatants were extracted, and protein concentration was measured using Pierce™ BCA Protein Assay Kit according to the manufacturer's protocol. A minimum of 10 µg of muscle extracts were subjected to blue-native gel electrophoresis using Native PAGE™ 4–16% Bis-Tris 1.0 mm Mini Protein Gels following the manufacturer's instructions before immobilising onto an Immobilon-P PVDF membrane for immunoblotting analysis.

### Statistics

No blinding or randomisation was applied in any of the experiments conducted in this study. Whenever feasible, experiments were performed in triplicate; however, limitations in available material occasionally prevented this. The precise number of replicates is detailed in each figure legend. Statistical significance was determined using the following indicators: *$P < 0.05$, **$P < 0.01$, and ***$P < 0.001$, with exact $p$ values provided in the figure legends.

## Data availability

Microarray data has been deposited in ArrayExpress (accession number E-MTAB-14251). Diagnostic genomic data were available from the authors on request, as due to consent agreement restrictions, patient diagnostic genomic data could not be made freely available and were therefore not deposited in a public database.

The source data of this paper are collected in the following database record: biostudies:S-SCDT-10_1038-S44321-024-00172-5.

## Peer review information

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

## Acknowledgements

JXT is supported by a PhD studentship funded by the Lily Foundation and a Newcastle University Overseas Research Scholarship. RWT is funded by the Wellcome Centre for Mitochondrial Research (203105/Z/16/Z), the Mitochondrial Disease Patient Cohort (UK) (G0800674), the Medical Research Council International Centre for Genomic Medicine in Neuromuscular Disease (MR/S005021/1), the Medical Research Council (MR/W019027/1),

the UK NIHR Biomedical Research Centre for Ageing and Age-related disease award to the Newcastle upon Tyne Foundation Hospitals NHS Trust, the Pathological Society, LifeArc and the UK NHS Highly Specialised Service for Rare Mitochondrial Disorders of Adults and Children. RWT, AP and MO receive additional support from the Mito Foundation and the Lily Foundation. LVH, PN, PP, CAP, PRG and MM are supported by core funding from UK Research and Innovation (UKRI) Medical Research Council (MRC), UK (MC_UU_00015/4 and MC_UU_00028/3), UKRI MRC award MC_PC_21046 to the National Mouse Genetics Network Mitochondria Cluster (MitoCluster), LifeArc, AFM Téléthon, The Champ, The Lily, Comini and CureMito Foundations. PP and MM were supported by the Marie Sklodowska-Curie ITN-REMIX grant (Grant 721757). Sequencing was performed in the Genomics Facility of the Cancer Research UK (CRUK) Cambridge Institute.

## Author contributions

**Lindsey Van Haute**: Conceptualisation; Data curation; Software; Formal analysis; Supervision; Validation; Investigation; Visualisation; Writing—original draft; Writing—review and editing. **Petra Páleníková**: Data curation; Software; Formal analysis; Validation; Investigation; Visualisation; Methodology; Writing—review and editing. **Jia Xin Tang**: Data curation; Formal analysis; Validation; Investigation; Visualisation; Writing—original draft; Writing—review and editing. **Pavel A Nash**: Investigation. **Mariella T Simon**: Data curation; Formal analysis; Validation; Investigation; Visualisation; Writing—review and editing. **Angela Pyle**: Investigation; Writing—review and editing. **Monika Oláhová**: Formal analysis; Investigation; Writing—review and editing. **Christopher A Powell**: Formal analysis; Investigation; Writing—review and editing. **Pedro Rebelo-Guiomar**: Formal analysis; Validation; Investigation; Writing—original draft; Writing—review and editing. **Alexander Stover**: Investigation; Writing—review and editing. **Michael Champion**: Resources; Writing—review and editing. **Charulata Deshpande**: Resources; Writing—review and editing. **Emma L Baple**: Resources; Writing—review and editing. **Karen L Stals**: Resources; Writing—review and editing. **Sian Ellard**: Resources; Writing—review and editing. **Olivia Anselem**: Resources; Writing—review and editing. **Clémence Molac**: Resources; Writing—review and editing. **Giulia Petrilli**: Resources; Writing—review and editing. **Laurence Loeuillet**: Resources; Writing—review and editing. **Sarah Grotto**: Resources; Writing—review and editing. **Tania Attie-Bitach**: Supervision; Project administration; Writing—review and editing. **Jose E Abdenur**: Conceptualisation; Supervision; Funding acquisition; Writing—review and editing. **Robert W Taylor**: Conceptualisation; Data curation; Formal analysis; Funding acquisition; Visualisation; Writing—original draft; Project administration; Writing—review and editing. **Michal Minczuk**: Conceptualisation; Data curation; Formal analysis; Supervision; Funding acquisition; Validation; Visualisation; Methodology; Writing—original draft; Project administration; Writing—review and editing.

Source data underlying figure panels in this paper may have individual authorship assigned. Where available, figure panel/source data authorship is listed in the following database record: biostudies:S-SCDT-10_1038-S44321-024-00172-5.

## Disclosure and competing interests statement

MM is a founder, shareholder and member of the Scientific Advisory Board of Pretzel Therapeutics, Inc. LVH is director of NextGenSeek Ltd. The remaining authors declare no competing interests.

# Expanded View Figures

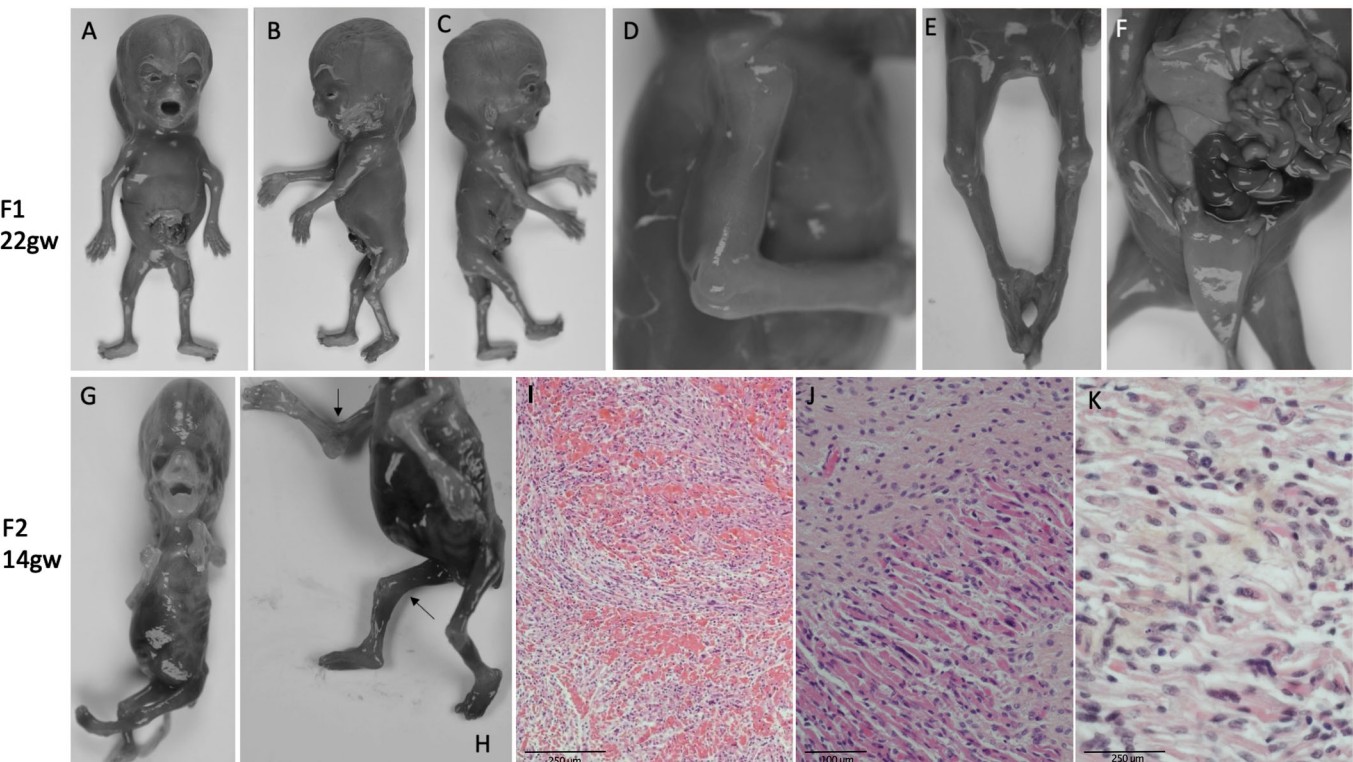

**Figure EV1. Images and muscle histology of Foetuses 4 and 5.**

Images and muscle histology of Foetus 4 (22 gw, A–F) and Foetus 5 (13,2 gw, G–K). (A–C) Face and profile pictures showing cystic hygroma, facial dysmorphisms with hypertelorism, high and large forehead and low-set ears. (D) Right arm showing elbow ankylosis. (E) Lower limbs showing bilateral knee ankylosis. (F) Autopsy showing pulmonary hypoplasia and distended bladder. (G, H) Face and profile pictures showing contractures of four limbs with arthrogryposis, multiple pterygia (colli, elbows, knees) and generalised muscular atrophy. (I–K) Histological analysis showing rare muscular fibres of irregular in size, with an excess of conjunctive tissue. Both foetuses present intra-uterine growth retardation and hydrops fetalis.

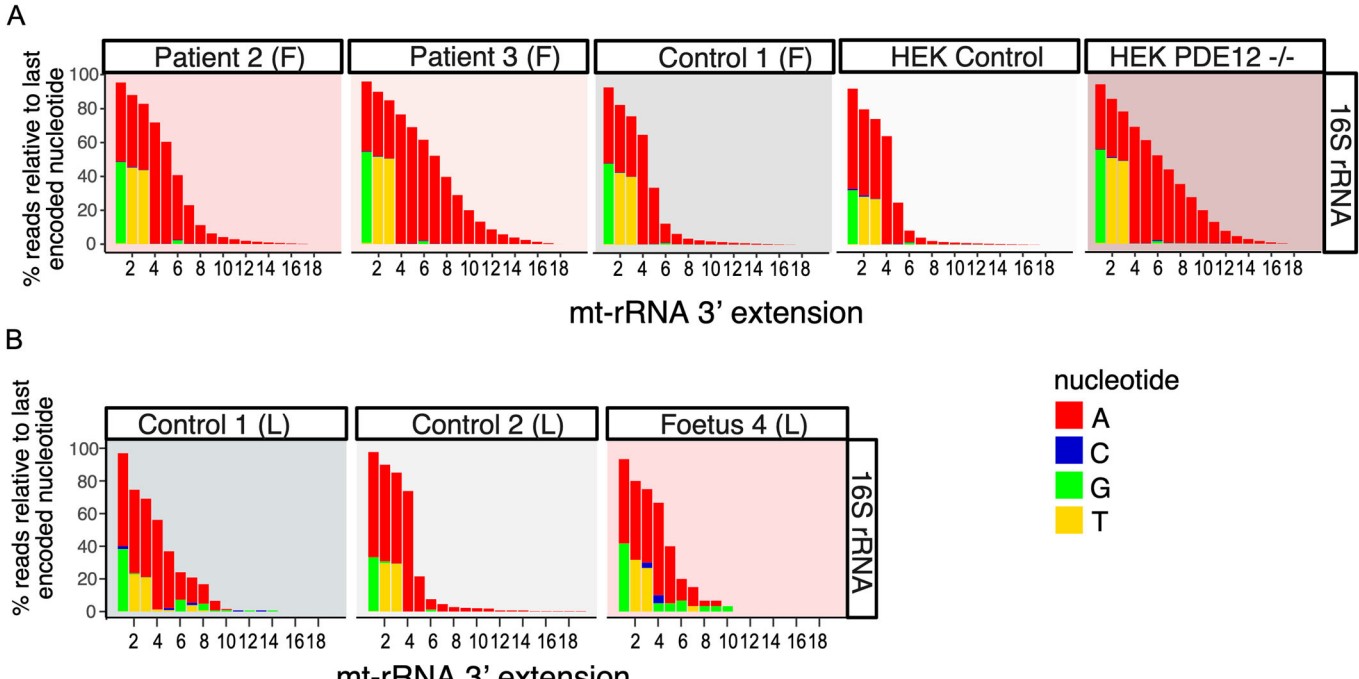

**Figure EV2. Spurious 3′ adenylation of 16S rRNA.**

(A) Representation of reads beyond the encoded nucleotides for 16S rRNA for Patient 2 and Patient 3 fibroblasts, fibroblast control, HEK PDE12 KO cells and control HEK cells ascertained by MPAT-Seq. ($n = 1$). (B) Representation of reads beyond the encoded nucleotides for 16S rRNA for RNA extracted from liver of Foetus 4 liver and age-matched control liver ascertained by MPAT-Seq. ($n = 1$). Source data are available online for this figure.

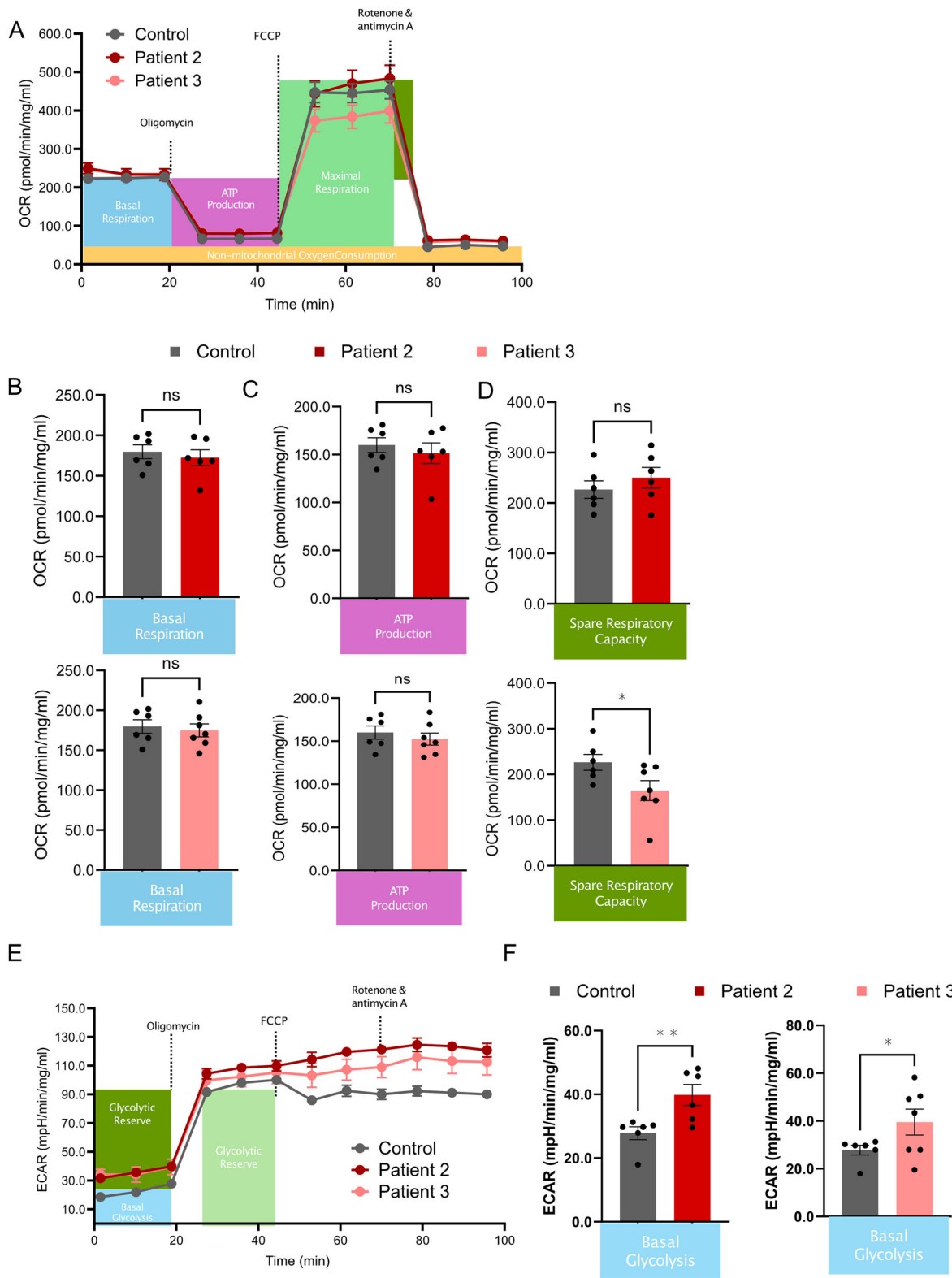

**Figure EV3. Mitochondrial respiration in patient fibroblasts.**

(A) Mitochondrial oxygen consumption rate (OCR) as measured by a flux analyser on control fibroblasts (grey) and fibroblasts of Patient 2 (red) and Patient 3 (pink). (B) Basal respiration was measured before the addition of Oligomycin. (C) Oligomycin was added to determine ATP production. (D) FCCP was added to determine the maximal respiration and spare respiratory capacity (*$p = 0.0263$). (E) Extracellular acidification rate. (F) Basal glycolysis. Error bars in (A–F) represent standard error of mean. Each measurement was conducted in at least three replicates (*$p = 0.0427$, **$p = 0.0055$). The one-tail $t$-test was used to calculate the $p$ values. Dots show individual measurements ($n = 6$, technical replicates).

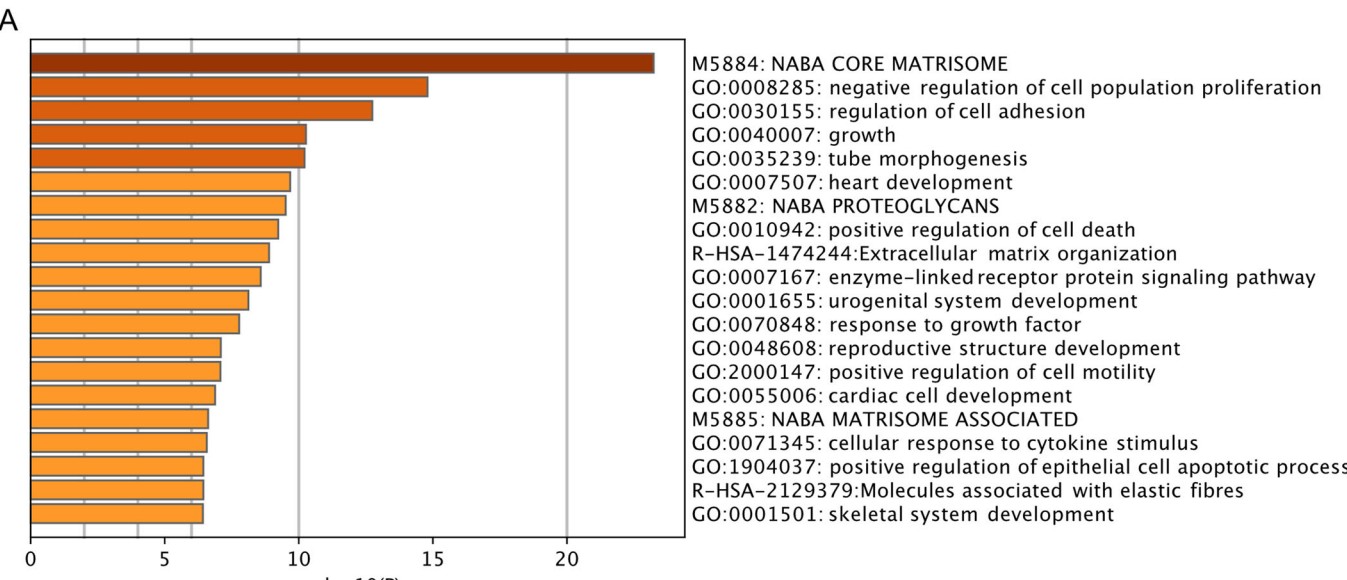

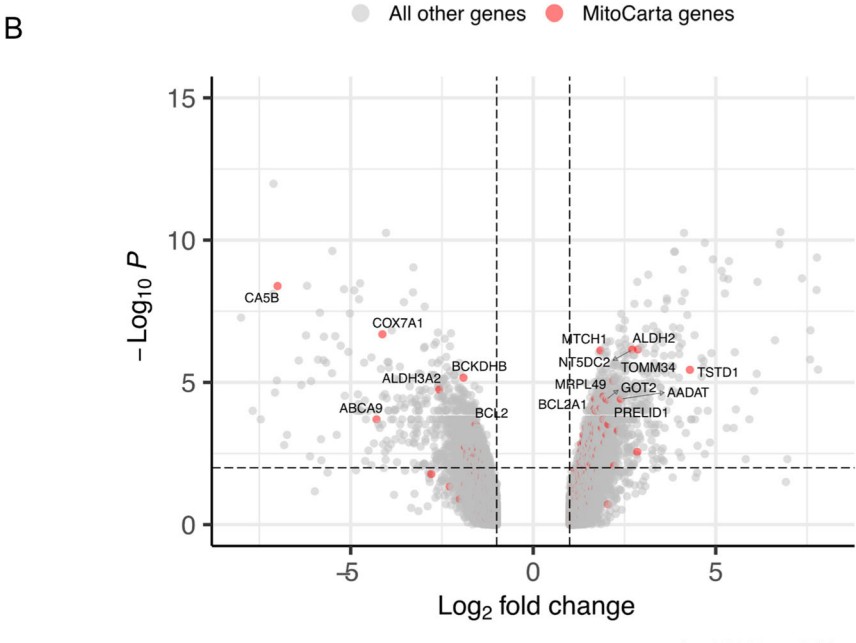

**Figure EV4.  Gene expression analysis in patient fibroblasts.**

(A) System-level analysis of Affymetrix Clariom D transcriptome arrays comparing the RNA levels in Patients 2 and 3 with unaffected controls. Statistical analysis was done using the ANOVA test and FDR-corrected values. The multiple-testing correction is based on the approach of Benjamini, Hochberg, and Yekutieli. (B) Volcano plot showing the differences in gene expression between RNA from Patients 2 and 3 compared with unaffected controls. Red dots represent genes that localise inside the mitochondria, according to MitoCarta, with the most significantly altered genes labelled. Statistical analysis as described for EV4A. ($n = 1$ biological replicate).

