## [Peer Review File · EMBO Molecular Medicine]

Pathogenic PDE12 variants impair mitochondrial RNA processing causing neonatal mitochondrial disease

Michal Minczuk, Lindsey Van Haute, Petra Páleníková, Jia Tang, Mariella Simon, Angela Pyle, Monika Oláhová, Christopher Powell, Pedro Rebelo-Guiomar, Alexander Stover, Michael Champion, Charu Deshpande, Emma Baple, Karen Stals, Sian Ellard, Olivia Anselem, Clémence Molac, Giulia Petrilli, Laurence Loeuillet, Sarah Grotto, Tania Attie-Bitach, Jose Abdenur, Robert Taylor, and Pavel Nash

Corresponding authors: Michal Minczuk (mam@mrc-mbu.cam.ac.uk) , Robert Taylor (Robert.Taylor@newcastle.ac.uk)

Review Timeline:

Submission Date:	5th Apr 24
Editorial Decision:	10th May 24
Revision Received:	5th Sep 24
Editorial Decision:	25th Sep 24
Revision Received:	18th Oct 24
Accepted:	29th Oct 24

Editor: Zeljko Durdevic

Transaction Report:

10th May 2024

Dear Prof. Taylor,

Thank you for the submission of your manuscript to EMBO Molecular Medicine. We have now received feedback from the three reviewers who agreed to evaluate your manuscript. All three referees recognize interest of the study but also raise important and partially overlapping concerns that should be addressed in a major revision. Particular attention should be given to providing more mechanistic insights into the biological effects of PDE12 mutations and mitochondrial disease pathogenesis. If you would like to discuss further the points raised by the referees, I am available to do so via email or video. Let me know if you are interested in this option.

We would welcome the submission of a revised version within three months for further consideration. Please let us know if you require longer to complete the revision.

I look forward to receiving your revised manuscript.

Yours sincerely,

Zeljko Durdevic

We require:

- 1) A .docx formatted version of the manuscript text (including legends for main figures, EV figures and tables). Please make sure that the changes are highlighted to be clearly visible.
- 2) Individual production quality figure files as .eps, .tif, .jpg (one file per figure). For guidance, download the 'Figure Guide PDF': (<https://www.embopress.org/page/journal/17574684/authorguide#figureformat>).
- 3) A .docx formatted letter INCLUDING the reviewers' reports and your detailed point-by-point responses to their comments. As part of the EMBO Press transparent editorial process, the point-by-point response is part of the Review Process File (RPF), which will be published alongside your paper.
- 4) A complete author checklist, which you can download from our author guidelines (<https://www.embopress.org/page/journal/17574684/authorguide#submissionofrevisions>). Please insert information in the checklist that is also reflected in the manuscript. The completed author checklist will also be part of the RPF.
- 5) Please note that all corresponding authors are required to supply an ORCID ID for their name upon submission of a revised manuscript.

6) It is mandatory to include a 'Data Availability' section after the Materials and Methods. Before submitting your revision, primary datasets produced in this study need to be deposited in an appropriate public database, and the accession numbers and database listed under 'Data Availability'. Please remember to provide a reviewer password if the datasets are not yet public (see <https://www.embopress.org/page/journal/17574684/authorguide#dataavailability>).

13) Author contributions: You will be asked to provide CRediT (Contributor Role Taxonomy) terms in the submission system. These replace a narrative author contribution section in the manuscript.

14) A Conflict of Interest statement should be provided in the main text.

Please also suggest a striking image or visual abstract to illustrate your article as a PNG file 550 px wide x 300-800 px high.

**** Reviewer's comments ****

Referee #1 (Remarks for Author):

This manuscript by Van Haute et al presents evidence that mutations in PDE12 underlie a severe fetal or neonatal presentation of mitochondrial disease, primarily impacting the oxidative phosphorylation machinery (OXPHOS). The authors have identified novel homozygous PDE12 mutations and show that they result in aberrant polyadenylation of mitochondrial mRNAs and subsequent reduction in stability of some OXPHOS-related proteins, in a tissue-specific manner. The authors combine genetic, biochemical, metabolic and transcriptomic approaches to present a compelling case for considering PDE12 mutations in cases of mitochondrial disease of unknown etiology.

Major comments:

The manuscript is, to a large extent, well-written and the conclusions drawn are justified on the basis of the data presented. There are, however, a couple of points that need addressing:

- 1) When the mutations are presented, the potential effects at the protein level are not sufficiently thought out. i.e. the Y155C mutation, authors should also consider the effect of substituting an amino acid with a free thiol group, which is reactive. In addition, the R41P change would potentially/likely disrupt the amphipathic helix of the MTS - stating this implicitly is much stronger than describing a generalized mitochondrial import problem.
- 2) There are visible differences in polyadenylation between the various mitochondrial tRNAs in the patients, but the authors make no mention of - or try to explain - why these differences might come about. This deserves some discussion with respect to patient phenotypes and potential molecular basis for the biological effects of the mutations.

Minor comments:

- 1) Some attention to grammar would help improve the flow in a few spots.
- 2) Several of the figure legends (Figures 3 - 6) require more information in order to better describe the results presented. For instance, the legend to Fig 4 should define MT-TE etc, since only those working specifically in the area of mitochondrial RNA metabolism would recognize those abbreviations. In addition, there should be a description of the statistics used in the analysis, especially given the asterisks that allude to statistical significance of the differences in spurious polyadenylation in the patient fibroblasts. It would also be beneficial if the title for figure 6 highlighted that the results were obtained from skeletal muscle.
- 3) Line 440 - mentions impacts on OXPHOS complexes, except for Complex II - but the Western blot in Fig 6C suggests differently. This requires comment.

Referee #2 (Comments on Novelty/Model System for Author):

Cell lines: Were all exon sequences of PDE12 gene in fibroblasts derived from patients and age-matched controls verified by Sanger sequencing to make sure these cell lines lacking functional variant? Otherwise, the biochemical data do not make sense.

Referee #2 (Remarks for Author):

Mitochondrial biogenesis requires the interplay between mitochondrial DNA (mtDNA) coding for 13 polypeptides for OXPHOS, 22 tRNAs and 2 rRNAs, and nuclear genes encoding approximately 1500 mitochondrial proteins including 72 OXPHOS subunits and RNA processing enzymes, which are synthesized in cytosol and imported into mitochondria. Impaired mitochondrial functions arising from defects in both mitochondrial and nuclear genomes have been associated with a wide spectrum of clinical presentations including neuromuscular disorders. In particular, an increasing number of families have been identified in which Mendelian genetic disorders implicating defective mitochondrial RNA metabolism. In this manuscript, authors identified three disease-causing PDE12 variants in three genetically unrelated families, which are associated with mitochondrial respiratory chain deficiencies and wide-ranging clinical presentations in utero and within the neonatal period with muscle and brain involvement leading to marked cytochrome c oxidase (COX) deficiency in muscle and severe lactic acidosis. Whole exome sequencing of affected probands revealed novel, segregating bi-allelic missense PDE12 variants affecting highly conserved residues. Patient-derived primary fibroblasts demonstrate diminished steady-state levels of PDE12 protein, whilst mitochondrial poly(A)-tail RNA sequencing (MPAT-Seq) revealed an accumulation of spuriously polyadenylated mitochondrial RNA species, consistent with perturbed function of PDE12 protein. Authors suggest that PDE12 regulates mitochondrial RNA processing in human tissues and that loss of PDE12 protein function results in neurological and muscular phenotypes.

This is an interesting study and worth to be published in EMBO Medicine. However, authors should address the following concerns before accepting for publication.

1. Introduction needs to be revised. "Nuclear genes encoding approximately 1500 mitochondrial proteins including 72 OXPHOS subunits and RNA processing enzymes" should be included. "two long polycistronic transcripts" should be described in detail (Ojala et al, 1981).

Montoya et al. Cell, 1983, 34,151-159; Xiao et al. 2020 48:11113-11129). "the tRNA punctuation model (Anderson et al., 4 97 1981; Ojala et al, 1981)" is not only model for mt-RNA processing, Guan et al. lab proposed the asymmetrical processing model for light strand-RNA precursors (Nucleic Acids Res, 2019 47:10340-10356. 2020 48:11113-11129). These should be included and cited. Anderson et al., 4 97 1981 did not described the tRNA punctuation model should not be cited.

2. The 5' and 3' end processing defects of mitochondrial RNA precursors due to mtDNA mutations should be discussed in this manuscript: 5' end processing defect (Wang et al. Circ Res. 2011;108:862-70; Zhao et al. Nucleic Acids Res, 2019 47:10340-10356.; Xiao et al 2020 48:11113-11129) and 3' end processing defect (Ji, et al., J Biol Chem. 2021;297:100816. Guan et al., Mol Cell Biol. 1998 18:5868-79).

3. Cell lines: Were all exon sequences of PDE12 gene in fibroblasts derived from patients and age-matched controls verified by Sanger sequencing to make sure these cell lines lacking functional variant? Otherwise, the biochemical data do not make sense.

4. Regarding the mitochondrial targeting sequence (MTS) for PDE12 protein, the mitoprot II (<ftp://ftp.biologie.ens.fr/pub/molbio>) program predicted the residue valine at N-terminal is the cleavage site of PDE12 protein. Author claimed that the p.Arg41Pro variant resided at MTS of PDE12. The mitochondrial localization experiments should be performed, as described at this group (Nucleic Acids Res. 2011, 39, 7755).

5. It is very common that severity of clinical and biochemical phenotypes were correlated with the altered structure and function caused by different variants. Here, the differences about the biochemical data among three variants should be discussed in detail. I suggested the merge of Figure 6 and supplemental figure 3.

6. Seahorse data for measuring the mitochondrial function in supplemental Figure 4 were not very convincing. To further evaluate the effect of PED12 variants on mitochondrial function, the in-gel activity with BN-PAGE or COX-SDH staining experiments should be performed (Jia et al. Nucleic Acids Res. 2022;50:9368-9381).

7. Minor issue: COXI should be changed to CO1.

Referee #3 (Comments on Novelty/Model System for Author):

Technical quality: Overall, the study seems to have been performed well, with 2 controls as comparison group and 2 or more independent samples expressing PDE12 variants; this latter aspect of the study provides for robustness of the results.

Weaknesses: 1) western blot quantification should have been better described in the legends; in particular, are the light grey bars meant to be standard deviation? 2) It is a pity that the pathway analysis was performed on fibroblasts, since the PDE12 variant fibros did not show decreased ETC complex subunit abundance (Suppl Fig3), though it is noted that another type of sample would have been difficult to obtain (and more so from both patients); 3) The bioenergetics (Suppl Fig4) should include statistical analysis. 4) Though the poly-A read counts are clearly higher in most PDE12 variant samples, it would be helpful to provide some type of statistical analysis.

Referee #3 (Remarks for Author):

This is a solid study (with a few weaknesses noted above), and the study is novel in 2 aspects: 1) first description of pathogenic variant in PDE12; 2) first description of a pathogenic variant in a mitoribosome quality control enzyme (which, in effect, demonstrates the importance of not only mitoribosome QC but also the particular poly-A removal function of PDE12. Where my

enthusiasm wavers is in the relatively narrow scope of the study. In particular, the study does not provide solid insights(into PDE12 function, or into mitochondrial disease pathogenesis more broadly) beyond the demonstration that the misense mutations are pathogenic.

Referee #1 (Remarks for Author):

This manuscript by Van Haute et al presents evidence that mutations in PDE12 underlie a severe fetal or neonatal presentation of mitochondrial disease, primarily impacting the oxidative phosphorylation machinery (OXPHOS). The authors have identified novel homozygous PDE12 mutations and show that they result in aberrant polyadenylation of mitochondrial mRNAs and subsequent reduction in stability of some OXPHOS-related proteins, in a tissue-specific manner. The authors combine genetic, biochemical, metabolic and transcriptomic approaches to present a compelling case for considering PDE12 mutations in cases of mitochondrial disease of unknown etiology.

Thank you for your positive and detailed assessment of our manuscript. We appreciate your recognition of our work and your thoughtful comments regarding the impact of PDE12 mutations on mitochondrial disease.

Major comments:

The manuscript is, to a large extent, well-written and the conclusions drawn are justified on the basis of the data presented. There are, however, a couple of points that need addressing:

- 1) When the mutations are presented, the potential effects at the protein level are not sufficiently thought out. i.e. the Y155C mutation, authors should also consider the effect of substituting an amino acid with a free thiol group, which is reactive.

Upon further inspection, we concluded that predicting the effect of the Y155C variant on thiol reactivity based solely on structure is challenging. PDE12 contains 16 cysteine residues, approximately half of which are exposed. However, residue 155 is only minimally exposed in the current structural models. Consequently, it appears that structural alterations have a more significant impact on PDE12 in the case of the Y155C variant. This assessment aligns with the substantial reduction in steady-state levels observed for the Y155C mutant protein (**Fig. 3**). We added a sentence to address this (**Page 10**)

In addition, the R41P change would potentially/likely disrupt the amphipathic helix of the MTS - stating this implicitly is much stronger than describing a generalized mitochondrial import problem.

In response to the Referee's comment, we conducted additional *in silico* predictions. According to AlphaFold 3 analyses, residue R41 is not located within a helix. The local RMSD between the WT and R41P structures is 0.131 Å, suggesting that the impact of the mutation is likely more functional than structural. Consequently, we decided to further investigate the involvement of R41 in pre-protein processing through experimental means. By comparing the mitochondrial processing of transiently overexpressed wild-type PDE12 with that of the PDE12 R41P variant, we now demonstrate that the latter is not processed, confirming a mitochondrial import issue (**Page 11** and **Fig. 3C**).

- 2) There are visible differences in polyadenylation between the various mitochondrial tRNAs in the patients, but the authors make no mention of - or try to explain - why these differences might come about. This deserves some discussion with respect to patient phenotypes and potential molecular basis for the biological effects of the mutations.

Differences among various mitochondrial tRNAs were also observed in our previous study (Pearce et al. 2017), and the underlying reasons for these discrepancies remain unclear. In response to the referee's comment, we now discuss this issue in the manuscript, emphasizing that the extent of PDE12 dysfunction may play a crucial role in determining which tRNA is affected (**Page 16**).

Minor comments:

- 1) Some attention to grammar would help improve the flow in a few spots. While we find it challenging to pinpoint the specific sections the Referee is referring to, we would like to assure you that the manuscript has been thoroughly reviewed by native English speakers to enhance its grammatical accuracy and overall flow.
- 2) Several of the figure legends (Figures 3 - 6) require more information in order to better describe the results presented. For instance, the legend to Fig 4 should define MT-TE etc, since only those working specifically in the area of mitochondrial RNA metabolism would recognize those abbreviations. Thank you for the suggestion. We have now defined the tRNA abbreviations, both in the figure legends and the manuscript text.

In addition, there should be a description of the statistics used in the analysis, especially given the asterisks that allude to statistical significance of the differences in spurious polyadenylation in the patient fibroblasts. It would also be beneficial if the title for figure 6 highlighted that the results were obtained from skeletal muscle. We have made the requested changes

Line 440 - mentions impacts on OXPHOS complexes, except for Complex II - but the Western blot in Fig 6C suggests differently. This requires comment. Western blot analysis shows no difference in steady-state level (or a slight upregulation) of SDHA in patient 2. Only a marginal reduction of SDHB in foetal individual 4 was observed. We have modified the text on **page 13** to address this issue.

Referee #2 (Comments on Novelty/Model System for Author):

Cell lines: Were all exon sequences of PDE12 gene in fibroblasts derived from patients and age-matched controls verified by Sanger sequencing to make sure these cell lines lacking functional variant? Otherwise, the biochemical data do not make sense.

We have verified the fibroblast cell lines from the patients and can confirm the presence of the variants, as evidenced by the Sanger sequencing results presented below. In response to the comment on the biochemical data, it is important to note that patient-derived fibroblasts often do not exhibit phenotypes of OXPHOS dysfunction, even in cases of mitochondrial pathology being present in other, usually post-mitotic, tissues. The primary reason for their routine use is their accessibility; thus, not analysing them would represent a missed opportunity in instances where a phenotype is present in patient-derived fibroblasts.

Patient 2

PDE12: ENSG00000174840 / NM_177966.6, NP_808881.3

Homozygous missense variant in exon 1 (c.464A>G, p.Tyr155Cys)

Forward sequencing

Reverse sequencing

Patient 3

PDE12: ENSG00000174840 / NM_177966.6, NP_808881.3

Homozygous missense variant in exon 1 (c.1115G>A, p.Gly372Glu)

Forward sequencing

Reverse sequencing

Referee #2 (Remarks for Author):

Mitochondrial biogenesis requires the interplay between mitochondrial DNA (mtDNA) coding for 13 polypeptides for OXPHOS, 22 tRNAs and 2 rRNAs, and nuclear genes encoding approximately 1500 mitochondrial proteins including 72 OXPHOS subunits and RNA processing enzymes, which are synthesized in cytosol and imported into mitochondria. Impaired mitochondrial functions arising from defects in both mitochondrial and nuclear genomes have been associated with a wide spectrum of clinical presentations including neuromuscular disorders. In particular, an increasing number of families have been identified in which Mendelian genetic disorders implicating defective mitochondrial RNA metabolism. In this manuscript, authors identified three disease-causing PDE12 variants in three genetically unrelated families, which are associated with mitochondrial respiratory chain deficiencies and wide-ranging clinical presentations in utero and within the neonatal period with muscle and brain involvement leading to marked cytochrome c oxidase (COX) deficiency in muscle and severe lactic acidosis. Whole exome sequencing of affected probands revealed novel, segregating bi-allelic missense PDE12 variants affecting highly conserved residues. Patient-derived primary fibroblasts demonstrate diminished steady-state levels of PDE12 protein, whilst mitochondrial poly(A)-tail RNA sequencing (MPAT-Seq) revealed an accumulation of spuriously polyadenylated mitochondrial RNA species, consistent with perturbed function of PDE12 protein. Authors suggest that PDE12 regulates mitochondrial RNA processing in human tissues and that loss of PDE12 protein function results in neurological and muscular phenotypes.

This is an interesting study and worth to be published in EMBO Medicine. Thank you for the positive feedback. We appreciate your recognition of our study's value for publication in *EMBO Molecular Medicine*.

However, authors should address the following concerns before accepting for publication.

1. Introduction needs to be revised. "Nuclear genes encoding approximately 1500 mitochondrial proteins including 72 OXPHOS subunits and RNA processing enzymes" should be included. "two long polycistronic transcripts" should be described in detail (Ojala et al, 1981. Montoya et al. Cell, 1983, 34,151-159; Xiao et al. 2020 48:11113-11129). "the tRNA punctuation model (Anderson et al., 4 97 1981; Ojala et al, 1981)" is not only model for mt-RNA processing, Guan et al. lab proposed the asymmetrical processing model for light strand-RNA precursors (Nucleic Acids Res, 2019 47:10340-10356. 2020 48:11113-11129). These should be included and cited. Anderson et al., 4 97 1981 did not described the tRNA punctuation model should not be cited.

We have changed the introduction to include the Referee's suggestions.

2. The 5' and 3' end processing defects of mitochondrial RNA precursors due to mtDNA mutations should be discussed in this manuscript: 5' end processing defect (Wang et al. Circ Res. 2011;108:862-70; Zhao et al. Nucleic Acids Res, 2019 47:10340-10356.; Xiao et al 2020 48:11113-11129) and 3' end processing defect (Ji, et al., J Biol Chem. 2021;297:100816. Guan et al., Mol Cell Biol. 1998 18:5868-79). We have added a section discussing the defects in 5' and 3' end processing caused by mutations in nuclear genes and mtDNA, along with the relevant references.

3. Cell lines: Were all exon sequences of PDE12 gene in fibroblasts derived from patients and age-matched controls verified by Sanger sequencing to make sure these cell lines lacking functional variant? Otherwise, the biochemical data do not make sense.

We have verified the patients' fibroblast cell lines and can confirm that the respective variants present in the cells we have studied (see above).

4. Regarding the mitochondrial targeting sequence (MTS) for PDE12 protein, the mitoprot II (<ftp://ftp.biologie.ens.fr/pub/molbio>) program predicted the residue valine at N-terminal is the cleavage site of PDE12 protein. Author claimed that the p.Arg41Pro variant resided at MTS of PDE12. The mitochondrial localization experiments should be performed, as described at this group (Nucleic Acids Res. 2011, 39, 7755).

In response to the referee's comment, we conducted additional in silico analyses. AlphaFold 3 predictions indicate that residue Arg41 is not located within a mitochondrial targeting sequence (MTS)-forming amphipathic helix. Consequently, we decided to further investigate the role of Arg41 in pre-protein processing through experimental methods. By comparing the mitochondrial processing of transiently overexpressed wild-type PDE12 with that of the PDE12 Arg41Pro variant, we demonstrate that the latter is not processed, confirming a mitochondrial import issue (**Page 11** and **Fig. 3C**). It is plausible that Arg41Pro disrupts the recognition site for mitochondrial processing peptidase (MPP), which is predicted to cleave between Cys42 and Val43 (**PMID: 12433926**).

5. It is very common that severity of clinical and biochemical phenotypes were correlated with the altered structure and function caused by different variants. Here, the differences about the biochemical data among three variants should be discussed in detail. I suggested the merge of Figure 6 and supplemental figure 3. Thank you for the suggestion. We have merged **Fig. 6** and **Supplementary Fig. 3**. In doing so, we link the severe phenotype observed in foetal individuals 4 and 5 to the significant reduction of functional PDE12 levels in the mitochondria due to import issues. In contrast, the other patients have low but detectable levels of PDE12, which likely result in a reduced yet existing activity of the remaining PDE12 protein.

6. Seahorse data for measuring the mitochondrial function in supplemental Figure 4 were not very convincing. To further evaluate the effect of PED12 variants on mitochondrial function, the in-gel activity with BN-PAGE or COX-SDH staining experiments should be performed (Jia et al. Nucleic Acids Res. 2022;50:9368-9381). Our findings clearly show that the effects of the PDE12 variants are more pronounced in skeletal muscle tissue compared to fibroblasts, as evidenced by Seahorse measurements, SDS-PAGE, and BN-PAGE analyses on fibroblast cells, which display only limited effects. Please refer to our comment above for additional context. Thus, we believe that conducting in-gel activity assays using BN-PAGE on fibroblasts would not yield additional insights regarding the function of the PDE12 variants.

1. Minor issue: COXI should be changed to CO1.
We changed this in the text and on the figures

Referee #3 (Comments on Novelty/Model System for Author):

Technical quality: Overall, the study seems to have been performed well, with 2 controls as comparison group and 2 or more independent samples expressing PDE12 variants; this latter aspect of the study provides for robustness of the results. Thank you for this positive evaluation of our work.

Weaknesses:

1) western blot quantification should have been better described in the legends; in particular, are the light grey bars meant to be standard deviation?

Thank you for this helpful comment. We have revised the figure legends to provide clearer explanations of what the bars represent.

2) It is a pity that the pathway analysis was performed on fibroblasts, since the PDE12 variant fibros did not show decreased ETC complex subunit abundance (Suppl Fig3), though it is noted that another type of sample would have been difficult to obtain (and more so from both patients);

We concur with the reviewer that conducting this analysis on muscle tissue would have been ideal. However, as the reviewer correctly pointed out, we were constrained by the limited amount of material available from both patients and controls.

3) The bioenergetics (Suppl Fig4) should include statistical analysis.

We have now added the statistical analysis for **Suppl Fig4 (now Figure EV3)**

4) Though the poly-A read counts are clearly higher in most PDE12 variant samples, it would be helpful to provide some type of statistical analysis.

We conducted a statistical analysis of polyA extension levels, as shown in **Figures 4 and 5**. Note: sample availability constraints meant that primarily one deep sequencing experiment was performed per sample, however, the observed differences across several mt-tRNAs support the robustness of this analysis.

Referee #3 (Remarks for Author):

This is a solid study (with a few weaknesses noted above), and the study is novel in 2 aspects:

1) first description of pathogenic variant in PDE12;

2) first description of a pathogenic variant in a mitoribosome quality control enzyme (which, in effect, demonstrates the importance of not only mitoribosome QC but also the particular poly-A removal function of PDE12. Where my enthusiasm wavers is in the relatively narrow scope of the study. In particular, the study does not provide solid insights(into PDE12 function, or into mitochondrial disease pathogenesis more broadly) beyond the demonstration that the misense mutations are pathogenic.

Thank you for your thoughtful evaluation and highlighting the novel aspects of our study. We appreciate your feedback on the study's scope and are motivated to explore PDE12 function and mitochondrial disease pathogenesis further in future research.

25th Sep 2024

Dear Dr. Minczuk,

Thank you for the submission of your revised manuscript to EMBO Molecular Medicine. I am pleased to inform you that we will be able to accept your manuscript pending the following final amendments:

1) In the main manuscript file, please do the following:

- Please address all comments suggested by our data editors listed below:

o Figure legends:

1. Please define the annotated p values ** as well as provide the exact p-values for the same in the legend of figure 3b; as appropriate.
 2. Please indicate the statistical test used for data analysis in the legends of figures EV 4a-b.
 3. Please note that the box plots need to be defined in terms of centre, bounds of box and percentile in the legends of figures 4c; 5b.
 4. Please note that information related to n is missing in the legends of figures 4c; EV 4b.
 5. Please note that n=2 in figure 3b.
 6. Please note that the error bars are not defined in the legends of figures EV 3a-e.
 7. Please note that scale bar and its definition are missing for figures EV 1i-k.
- Add callouts for individual panels for Fig 1, Fig 5, Fig EV1, EV3, EV4. Add callouts for Fig 4A.
- In Methods, add a statistical paragraph that should reflect all information that you have filled in the Authors Checklist, especially regarding randomization, blinding, replication.
- Indicate in legends exact n and exact p values, not a range, along with the statistical test used. To keep the figures "clear" some authors found providing an Appendix table Sx with all exact p-values preferable. You are welcome to do this if you want to.
- Please include structured Methods section that includes a Reagents and Tools Table (should be uploaded as a separate file) followed by a Methods and Protocols section. More information on how to adhere to this format as well as downloadable templates (.docx) for the Reagents and Tools Table can be found in our author guidelines:
<https://www.embopress.org/page/journal/17574684/authorguide#structuredmethods>
An example of a paper with Structured Methods can be found here:
<https://www.embopress.org/doi/full/10.1038/s44320-024-00037-6#sec-4>
- In Methods, provide the statement that in addition to the WMA Declaration of Helsinki the experiments also conformed to the principles set out in the Department of Health and Human Services Belmont Report.
- Author contributions: Please remove it from the manuscript and specify author contributions in our submission system. CRediT has replaced the traditional author contributions section because it offers a systematic machine-readable author contributions format that allows for more effective research assessment. You are encouraged to use the free text boxes beneath each contributing author's name to add specific details on the author's contribution. More information is available in our guide to authors:
<https://www.embopress.org/page/journal/17574684/authorguide#authorshipguidelines>
- Move data availability statement to the end of "Methods" section. Please remove the sentence "The authors confirm that the data supporting the findings of this study are available within the article and on request."
- 2) Appendix: Please add table of content on the first page and rename "Appendix" to "Appendix Supplementary Information" and update the callout in the text.
 - 3) Funding: Please make sure that information about all sources of funding are complete in both our submission system and in the manuscript. Complete funding information should be listed in our submission system, with funders and project numbers entered into main funding section (not the comments).
 - 4) Synopsis:
 - Synopsis image: Please format the image to 550 px-wide x (250-400)-px high and upload it as a high-resolution JPEG file.
 - Synopsis text: Please remove it from the main manuscript file and upload it as a separate .doc file.
 - Please check your synopsis text and image before submission with your revised manuscript. Please be aware that in the proof stage minor corrections only are allowed (e.g., typos).
 - 5) Source data: Please upload one (zip) file per figure.
 - 6) As part of the EMBO Publications transparent editorial process initiative (see our Editorial at <http://embomolmed.embopress.org/content/2/9/329>), EMBO Molecular Medicine will publish online a Review Process File (RPF) to accompany accepted manuscripts. This file will be published in conjunction with your paper and will include the anonymous referee reports, your point-by-point response and all pertinent correspondence relating to the manuscript. Let us know whether you agree with the publication of the RPF and as here, if you want to remove or not any figures from it prior to publication. Please note that the Authors checklist will be published at the end of the RPF.
 - 7) Please provide a point-by-point letter INCLUDING my comments as well as the reviewer's reports and your detailed responses (as Word file).

I look forward to reading a new revised version of your manuscript as soon as possible.

Yours sincerely,

Zeljko Durdevic

*** Instructions to submit your revised manuscript ***

1) a .docx formatted version of the manuscript text (including Figure legends and tables)

2) Separate figure files*

3) supplemental information as Expanded View and/or Appendix. Please carefully check the authors guidelines for formatting Expanded view and Appendix figures and tables at <https://www.embopress.org/page/journal/17574684/authorguide#expandedview>

4) a letter INCLUDING the reviewer's reports and your detailed responses to their comments (as Word file).

5) The paper explained: EMBO Molecular Medicine articles are accompanied by a summary of the articles to emphasize the major findings in the paper and their medical implications for the non-specialist reader. Please provide a draft summary of your article highlighting

This may be edited to ensure that readers understand the significance and context of the research.

Please refer to any of our published articles for an example.

6) Author contributions: the contribution of every author must be detailed in a separate section.

7) EMBO Molecular Medicine now requires a complete author checklist

(<https://www.embopress.org/page/journal/17574684/authorguide>) to be submitted with all revised manuscripts. Please use the checklist as guideline for the sort of information we need WITHIN the manuscript. The checklist should only be filled with page numbers where the information can be found. This is particularly important for animal reporting, antibody dilutions (missing) and exact values and n that should be indicated instead of a range.

8) Every published paper now includes a 'Synopsis' to further enhance discoverability. Synopses are displayed on the journal webpage and are freely accessible to all readers. They include a short stand first (maximum of 300 characters, including space) as well as 2-5 one sentence bullet points that summarise the paper. Please write the bullet points to summarise the key NEW findings. They should be designed to be complementary to the abstract - i.e. not repeat the same text. We encourage inclusion of key acronyms and quantitative information (maximum of 30 words / bullet point). Please use the passive voice. Please attach these in a separate file or send them by email, we will incorporate them accordingly.

You are also welcome to suggest a striking image or visual abstract to illustrate your article. If you do please provide a jpeg file 550 px-wide x 300-600px high.

9) A Conflict of Interest statement should be provided in the main text

10) Please note that we now mandate that all corresponding authors list an ORCID digital identifier. This takes <90 seconds to complete. We encourage all authors to supply an ORCID identifier, which will be linked to their name for unambiguous name identification.

Currently, our records indicate that the ORCID for your account is 0000-0001-8242-1420.

Link Not Available

11) Include a Reagents and Tools Table as part of the Methods section, which can be downloaded from our author guidelines (<https://www.embopress.org/page/journal/17574684/authorguide#structuredmethods>)

Photos 400-800 DPI

*Additional important information regarding figures and illustrations can be found at <https://bit.ly/EMBOPressFigurePreparationGuideline>. See also figure legend preparation guidelines: <https://www.embopress.org/page/journal/17574684/authorguide#figureformat>

***** Reviewer's comments *****

Referee #2 (Remarks for Author):

Authors adequately addressed the concerns raised by reviewers

Referee #3 (Comments on Novelty/Model System for Author):

Authors responded fully and satisfactorily to my concerns. Furthermore, I read Authors' replies to the other Reviewers, and re-read the manuscript: this is an improved and highly interesting manuscript and study.

Referee #3 (Remarks for Author):

Thank you for replying fully and satisfactorily to my concerns. Congratulations on a very interesting and clearly communicated study.

The authors addressed the remaining editorial issues.

Dear Dr. Durdevic,

We were very happy to learn that EMBO Molecular Medicine is willing to proceed with our manuscript. We appreciate your guidance throughout this process and look forward to seeing our work in print.

Thank you for your letter regarding the final amendments to our manuscript submission. I would like to confirm that I have addressed all the requested changes, as indicated below.

Best regards,

Michal

Dear Dr. Minczuk,

Thank you for the submission of your revised manuscript to EMBO Molecular Medicine. I am pleased to inform you that we will be able to accept your manuscript pending the following final amendments:

1) In the main manuscript file, please do the following:

- Please address all comments suggested by our data editors listed below:

o Figure legends:

1. Please define the annotated p values ** as well as provide the exact p-values for the same in the legend of figure 3b; as appropriate.

Done

2. Please indicate the statistical test used for data analysis in the legends of figures EV 4a-b.

Done

3. Please note that the box plots need to be defined in terms of centre, bounds of box and percentile in the legends of figures 4c; 5b.

Done

4. Please note that information related to n is missing in the legends of figures 4c; EV 4b.

Done

5. Please note that n=2 in figure 3b.

Done

6. Please note that the error bars are not defined in the legends of figures EV 3a-e.

Done

7. Please note that scale bar and its definition are missing for figures EV 1i-k.

Done

- Add callouts for individual panels for Fig 1, Fig 5, Fig EV1, EV3, EV4. Add callouts for Fig 4A.

Done

- In Methods, add a statistical paragraph that should reflect all information that you have filled in the Authors Checklist, especially regarding randomization, blinding, replication.

Done

- Indicate in legends exact n and exact p values, not a range, along with the statistical test used. To keep the figures "clear" some authors found providing an Appendix table Sx with all exact p-values preferable. You are welcome to do this if you want to.

In the figure legends

- Please include structured Methods section that includes a Reagents and Tools Table (should be uploaded as a separate file) followed by a Methods and Protocols section. More information on how to adhere to this format as well as downloadable templates (.docx) for the Reagents and Tools Table can be found in our author guidelines: <https://www.embopress.org/page/journal/17574684/authorguide#structuredmethods>
An example of a paper with Structured Methods can be found here:

<https://www.embopress.org/doi/full/10.1038/s44320-024-00037-6#sec-4>

Done

- In Methods, provide the statement that in addition to the WMA Declaration of Helsinki the experiments also conformed to the principles set out in the Department of Health and Human Services Belmont Report.

Done

- Author contributions: Please remove it from the manuscript and specify author contributions in our submission system. CRediT has replaced the traditional author contributions section because it offers a systematic machine-readable author contributions format that allows for more effective research assessment. You are encouraged to use the free text boxes beneath each contributing author's name to add specific details on the author's contribution. More information is available in our guide to authors:

<https://www.embopress.org/page/journal/17574684/authorguide#authorshipguidelines>

Done

- Move data availability statement to the end of "Methods" section. Please remove the sentence "The authors confirm that the data supporting the findings of this study are available within the article and on request."

Done

2) Appendix: Please add table of content on the first page and rename "Appendix" to "Appendix Supplementary Information" and update the callout in the text.

Done

3) Funding: Please make sure that information about all sources of funding are complete in both our submission system

and in the manuscript. Complete funding information should be listed in our submission system, with funders and project numbers entered into main funding section (not the comments).

To do

4) Synopsis:

- Synopsis image: Please format the image to 550 px-wide x (250-400)-px high and upload it as a high-resolution JPEG file.

Done

- Synopsis text: Please remove it from the main manuscript file and upload it as a separate .doc file.

Done

Done

5) Source data: Please upload one (zip) file per figure.

Done

6) As part of the EMBO Publications transparent editorial process initiative (see our Editorial at <http://embomolmed.embopress.org/content/2/9/329>), EMBO Molecular Medicine will publish online a Review Process File (RPF) to accompany accepted manuscripts. This file will be published in conjunction with your paper and will include the anonymous referee reports, your point-by-point response and all pertinent correspondence relating to the manuscript. Let us know whether you agree with the publication of the RPF and as here, if you want to remove or not any figures from it prior to publication. Please note that the Authors checklist will be published at the end of the RPF.

Agree

7) Please provide a point-by-point letter INCLUDING my comments as well as the reviewer's reports and your detailed responses (as Word file).

This document

I look forward to reading a new revised version of your manuscript as soon as possible.

Yours sincerely,

Zeljko Durdevic

***** Reviewer's comments *****

Referee #2 (Remarks for Author):

Authors adequately addressed the concerns raised by reviewers

Thank you for the feedback from Referee #2. We appreciate the positive acknowledgment.

Referee #3 (Comments on Novelty/Model System for Author):

Authors responded fully and satisfactorily to my concerns. Furthermore, I read Authors' replies to the other Reviewers, and re-read the manuscript: this is an improved and highly interesting manuscript and study.

Referee #3 (Remarks for Author):

Thank you for replying fully and satisfactorily to my concerns. Congratulations on a very interesting and clearly communicated study.

Thank you for the feedback from Referee #3. We're grateful for the positive comments and glad our responses addressed the concerns raised.

29th Oct 2024

Dear Prof. Minczuk,

We are pleased to inform you that your manuscript is accepted for publication and is now being sent to our publisher to be included in the next available issue of EMBO Molecular Medicine.
